# Polarity reversal of stable microtubules during neuronal development

Malina K. Iwanski, Albert K. Serweta, Jasper van Schelt, H. Noor Verwei, Bronte C. Donders and Lukas C. Kapitein*

## ABSTRACT

Long-distance transport in neurons relies on motor proteins that can move towards either the plus- or minus-end of microtubules. In axons, microtubules uniformly have a plus-end-out orientation, whereas dendrites of vertebrate neurons contain mixed polarity bundles: stable microtubules are typically minus-end-out, and dynamic microtubules are plus-end-out. This organization supports selective transport, yet how this dedicated microtubule organization is established is unclear. Here, we use single-molecule localization microscopy, expansion microscopy and live-cell imaging to examine how the microtubule cytoskeleton is reorganized during neuronal development in cultured rat hippocampal neurons. We find that early neurites contain mixed polarity microtubules, with stable microtubules initially mostly plus-end-out and often connected to centrioles. As neurons mature, these microtubules detach, slide and gradually reorient to become predominantly minus-end-out within the future dendrites. Moreover, prior to axon specification, neurons often have one or two minor neurites with an almost uniformly plus-end-out microtubule network. Our findings show how reorganization of stable microtubules underlies the establishment of the characteristic microtubule network in mature vertebrate neurons.

KEY WORDS: Microtubules, Neuronal polarity, Super-resolution microscopy

## INTRODUCTION

For their proper function, many cells in our body must be able to establish and maintain a polarized organization. Of these cells, one of the most exquisite examples is neurons, branched cells that bear both structurally and functionally distinct processes known as axons and dendrites. Information is received via the dendrites and integrated, after which an output is generated that is transmitted to other cells via the axon. For this functional distinction, these compartments also require a highly asymmetric distribution of specific molecular players. Thus, which cargo is transported into and out of these processes must be closely regulated. This long-distance cargo transport is effectuated by motor proteins of the kinesin superfamily and dynein, which move cargoes

Cell Biology, Neurobiology and Biophysics, Department of Biology, Faculty of Science, Utrecht University, Utrecht, Paduallaan 8, 3584 CH, The Netherlands.

*Author for correspondence (l.kapitein@uu.nl)

M.K.I., 0000-0002-4903-9796; H.N.V., 0009-0001-6169-7572; B.C.D., 0009-0004-3840-5529; L.C.K., 0000-0001-9418-6739

unidirectionally, helping to specifically localize them to one or the other compartment. Kinesins and dynein do so by walking along microtubules, intrinsically polar polymers that can grow to many micrometres in length. Microtubules are comprised of α,β-tubulin heterodimers that polymerize head-to-tail to form filaments with a distinct highly dynamic plus-end and a lesser dynamic and often capped minus-end. Most members of the kinesin superfamily walk towards the microtubule plus-end, whereas dynein walks towards the minus-end. In the axons of vertebrate neurons, microtubules are uniformly oriented with their plus-ends outwards away from the soma, whereas dendrites have a mixed microtubule array (Baas et al., 1988). This means that in the axon, most kinesin motors act as anterograde motors, carrying cargo away from the soma, whereas dynein serves as a retrograde motor. In the dendrites, however, both kinesins and dynein can theoretically act as anterograde and retrograde transporters, and additional layers of regulation are required to properly direct these motors. For example, microtubule-associated proteins (MAPs) have distinct activating and/or inhibitory effects on different motor proteins; some of these MAPs are specifically localized to dendrites and might thus help guide this transport (Karasmanis et al., 2018; Monroy et al., 2020). It has also been demonstrated that motors prefer microtubules bearing different tubulin post-translational modification (PTMs), with kinesin-1, for example, preferring microtubules marked by acetylation and detyrosination (typical of long-lived, stable microtubules) and kinesin-3 preferring microtubules marked by tyrosination (typical of labile microtubules with rapid turnover) (Cai et al., 2009; Tas et al., 2017).

Neurons are not directly established with this spectacularly asymmetric array of microtubules properly decorated by MAPs and PTMs. Instead, it is built as the cells polarize during development. The development of dissociated embryonic hippocampal neurons in culture is classically described in five stages (Fig. 1A) (Dotti et al., 1988). First, once the globular cells in suspension adhere to the coverslip, they extend a lamellipodium, a large protrusion with an extensive branched actin network (stage 1). Then, the cells develop minor processes called neurites, which grow and shrink (stage 2). One of these neurites is then specified to become the axon, growing out rapidly and persistently to become much longer than the rest (stage 3). Next, the remaining neurites also branch and develop further to adopt a dendritic identity (stage 4). Finally, the neurons form mature synapses with one another and are fully developed (stage 5). Concomitantly with this process, the microtubule cytoskeleton also adopts a highly polar architecture (Fig. 1A). In stage 1, the centrosome is still active, and the cells have a largely radial microtubule array; however, the centrosome is inactivated later during the developmental process (Stiess et al., 2010). Furthermore, based on the tracking of end-binding (EB) protein comets at the growing plus-ends of microtubules, it is known that neurites in stage 2 and 3 neurons contain microtubules of mixed

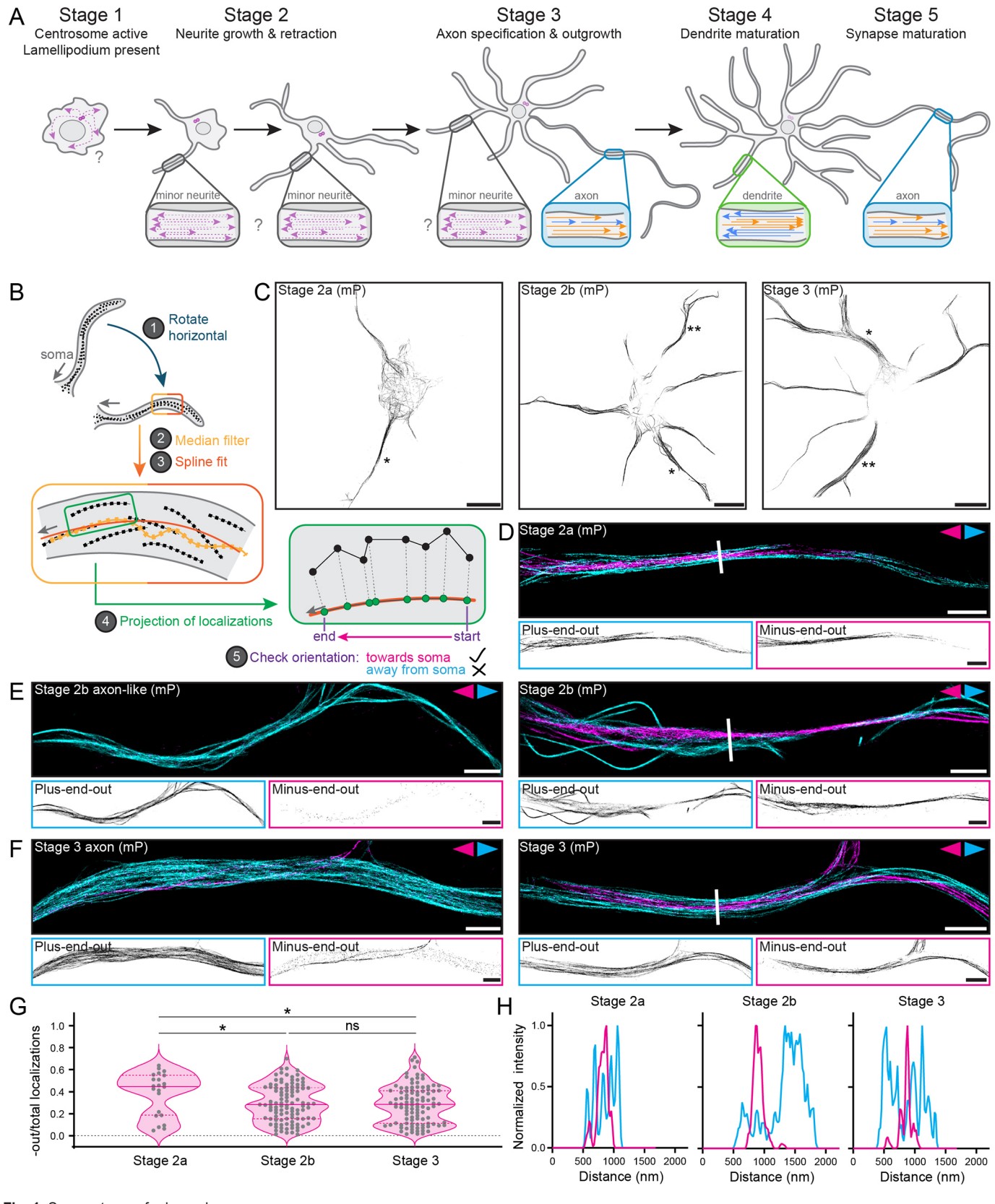

**Fig. 1.** See next page for legend.

Journal of Cell Science

orientation (Yau et al., 2016), but their precise organization remains unclear. In stage 4 and 5 neurons, it is known that axons contain uniformly plus-end-out microtubules, regardless of whether these

microtubules are stable or labile (Baas et al., 1988). In the dendrites, however, stable microtubules are preferentially oriented minus-end-out and enriched centrally, whereas dynamic or labile microtubules

**Fig. 1. Microtubules of opposite orientation are segregated from early on in neuronal development**. (A) Schematic showing the stages of neuronal development and the accompanying changes in the microtubule cytoskeleton. While it is known that microtubules are of mixed polarity (indicated by arrowheads) in minor neurites early on in development, the localization and orientation of stable (orange) and labile (blue) microtubules are unclear. (B) Schematic showing the steps of the analysis pipeline that allows for pseudo-colouring of tracks based on whether they are oriented towards (magenta) or away from (cyan) the soma. (C) Total tracks after filtering (all orientations) of representative neurons in early stage 2 (stage 2a), late stage 2 (stage 2b) and stage 3 imaged using motor-PAINT (mP). Scale bars: 10 μm. Single asterisks indicate neurites of corresponding stage shown in D and on the right in E and F. Double asterisks indicate neurites shown on the left in E and F. (D) A motor-PAINT reconstruction of a neurite from the stage 2a cell in C with microtubules pseudo-coloured based on whether their plus-end is oriented towards (magenta) or away from (cyan) the soma. Single-channel images are shown below. Scale bars: 2 μm. (E) Motor-PAINT reconstructions of two neurites from the stage 2b cell in C with microtubules pseudo-coloured based on whether their plus-end is oriented towards (magenta) or away from (cyan) the soma. Single-channel images are shown below. Scale bars: 2 μm. (F) Motor-PAINT reconstructions of two neurites from the stage 3 cell in C with microtubules pseudo-coloured based on whether their plus-end is oriented towards (magenta) or away from (cyan) the soma. Single-channel images are shown below. Scale bars: 2 μm. White lines in D,E,F indicate sections analysed in H. (G) Quantification of the fraction of localizations constituting minus-end-out tracks over the total amount of localizations for neurons in stages 2a, 2b and 3. Each dot represents one neurite. Medians (solid line; 0.45, 0.28, 0.29 for stages 2a, 2b and 3, respectively) and interquartile ranges (dashed lines; 0.19–0.55, 0.15–0.44, 0.11–0.41 for stages 2a, 2b and 3, respectively) are shown. $n$=22, 105, 103 neurites from $N$=9, 18, 17 cells for stages 2a, 2b and 3, respectively. Groups compared by two-tailed Mann–Whitney test (ns, not significant; *$P$≤0.05). (H) Intensity profiles of minus-end-out (magenta) and plus-end-out (cyan) microtubules along the white lines across the neurites shown in D, E (right) and F (right). Normalization was performed independently for the two orientations using the minimum and maximum values of those data sets.

are preferentially oriented plus-end-out and enriched peripherally (Katrukha et al., 2021; Tas et al., 2017). This raises the question: how does this organization emerge? Specifically, are microtubules in the minor neurites of stage 2 and 3 neurons already organized into bundles of preferred polarity as is observed in mature dendrites (Tas et al., 2017)? Do stable microtubules in these minor neurites have a preferred minus-end-out orientation as is observed in mature dendrites (Tas et al., 2017) and as has been suggested for developing neurons (Yau et al., 2016)? What are the effects of centrosome inactivation, and how do microtubules transition from the radial array to help form the first emerging neurites as neurons transition from stage 1 to stage 2?

Here, we address these questions using motor-PAINT to analyse the orientation of microtubules at different stages during neuronal development. Motor-PAINT is a single-molecule localization technique in which purified kinesin-1 motors are allowed to move across the microtubule cytoskeleton of permeabilized and semi-fixed cells to provide both a super-resolved image of the microtubule network and the orientation of these filaments based on the direction of movement of the kinesin motors (Tas et al., 2017). Importantly, although kinesin-1 has been shown to selectively move on stable, highly modified microtubules in cells (Cai et al., 2009; Reed et al., 2006), this is not the case after motor-PAINT sample preparation (Tas et al., 2017). This technique is particularly suitable because it provides insight into the organization of all microtubules, unlike EB comet tracking (Stepanova et al., 2003), which selectively reveals the orientation of growing microtubules. In addition, it has a much higher throughput than the electron microscopy-based hook decoration method (Baas et al., 1988). Our work revealed that, whereas some

neurites in stage 2 neurons contain almost exclusively plus-end-out microtubules, most emergent neurites contain both minus-end-out and plus-end-out microtubules, with microtubules of opposite orientation being segregated very early on. Furthermore, expansion microscopy revealed that, much like in mature dendrites, stable and labile microtubules (marked by acetylation and tyrosination, respectively) also form segregated networks from early in development. Stable microtubules are more minus-end-out oriented than the total population of microtubules from later in stage 2 onwards, but interestingly, they are preferentially plus-end-out oriented in early stage 2. This suggests that stable microtubules are initially nucleated at the centrosome and then released during stage 2, allowing them to reverse their orientation once they have a free minus-end. Consistent with this, expansion microscopy revealed stable microtubules emanating from the centrioles in stage 1 neurons, whereas in stage 2 and 3 neurons, we mostly observed short remnants of stable microtubules near the centrioles. Finally, we performed timelapse imaging using StableMARK, a recently established live-cell marker for stable microtubules (Jansen et al., 2023), and observed sliding and polarity reversal of stable microtubules during stage 2. Together, this work provides insights into the reorganization of the microtubule cytoskeleton during neuronal development and has implications for how the axonal and dendritic identities are established.

## RESULTS

### An optimized motor-PAINT protocol aids reconstruction quality and data interpretation

To robustly analyse microtubule orientation in developing neurons, optimization of the original motor-PAINT protocol (Tas et al., 2017) was necessary (Fig. S1A,B). Some of these adjustments have also been implemented in our recent motor-PAINT studies using MINFLUX and lattice light-sheet microscopy (Deguchi et al., 2023; Iwanski et al., 2023). First, we noticed that microtubules in young neurons sometimes wobbled during acquisitions, blurring the resulting reconstructions. To combat this, we additionally added a small amount (0.04%) of glutaraldehyde during the fixation step. Furthermore, we avoided the transfection of neurons with fluorescently labelled tubulin (e.g. mCherry–tubulin) as this requires electroporation in young neurons and limits the number of cells that can be imaged. Instead, we incubated neurons with phalloidin to allow us to easily find cells and concomitantly facilitate the identification of their developmental stage. During this step, we also added fluorescently labelled beads to allow for rapid and efficient drift correction. To improve localization precision and track length, we switched to a SNAP-tagged kinesin motor (based on *Drosophila* Kinesin heavy chain; DmKHC–SNAP), allowing us to take advantage of bright, photostable dyes such as Janelia Fluor (JF) 646. In addition, this motor was added in bulk rather than locally, which helped minimize microtubule wiggling during acquisitions (due to a lower local concentration of motors) and allowed us to seal our imaging chambers. This sealing greatly increased the lifetime of the assays and thus our throughput, likely by minimizing oxygen exposure, which in turn helps limit the acidification of the buffer by the glucose oxidase–catalase oxygen scavenger system that could otherwise reduce buffer quality such that motors are no longer able to walk well.

We also optimized the analysis and visualization of our results to improve reconstruction quality in the dense microtubule arrays of the neurites and simplify the interpretation of the resulting reconstructions (see Materials and Methods). In addition to the robust drift correction facilitated by the inclusion of beads, we also improved

the reconstructions by using TrackMate (Tinevez et al., 2017) to localize and track the kinesin motors, as it has tracking algorithms (e.g. the overlap tracker) that are more suitable for the dense microtubule networks in neurites. Furthermore, we adjusted our filtering to increase the number of tracks retained (and thereby the coverage of the microtubule network), most notably by carefully and locally filtering for angles between track 'steps' to remove only portions of the tracks with highly divergent angles signifying paused motors or reversals in direction resulting from erroneously linked spots (see Materials and Methods). Additionally, we predominantly colour-coded tracks not by their direction in the field of view, but rather by whether they were oriented towards or away from the soma (Fig. 1B; see Fig. S1B–E for examples). This orientation assignment was performed as follows: after selecting each neurite, it was rotated to be approximately horizontal to reduce the number of instances in which a track contained multiple *y*-values for a given *x*-value. To produce a midline for a given neurite or branch, the localizations from all the tracks within this neurite or branch were then treated as a whole, median filtered and subjected to spline fitting. After determining which end of this midline was closer to the soma, the localizations of each track were projected onto this midline, and the start and end points of the track were compared to determine whether the track was moving towards or away from the soma along the midline. This was more reliable than directly comparing whether the start or end point of the track was closer to the soma, especially in instances where neurites or their branches were curled back towards the soma.

## Microtubules are segregated by orientation from early on in development

With these improvements, we could reliably reconstruct the microtubule networks in neurons throughout their development, focusing on the period ∼5–75 h after plating (Fig. 1C; Fig. S1C–E). These neurons were predominantly in stages 1 to 4, with most neurons in stage 1 or 2 at earlier time points and stage 2 to 4 at later time points. We chose to focus on neurons in stages 2 and 3, but we subdivided those in stage 2 into early (stage 2a) and late (stage 2b) depending on the number and length of the neurites to better describe the changes in microtubule organization occurring early in development (Fig. 1A; see Materials and Methods). Most neurites had microtubules of both orientations, with microtubules of opposite orientation segregated (i.e. forming two spatially separated populations) across the width of these neurites, including those of stage 2a neurons, which were most commonly found a mere 5 h after plating [Fig. 1D,E (right),F (right), G,H]. Importantly, only some neurites had minus-end-out microtubules enriched centrally and plus-end-out microtubules enriched peripherally, as has been observed in mature dendrites (Tas et al., 2017). Instead, many neurites simply had adjacent populations of microtubules of opposite orientation [Fig. 1D,E (right), F (right),H]. Although we observed a wide spread in the fraction of minus-end-out microtubules per neurite at all developmental stages, neurites in stage 2b and stage 3 neurons had on average fewer minus-end-out microtubules than those of stage 2a neurons, with median values of 0.45, 0.28 and 0.29 for stage 2a, stage 2b and stage 3 neurons, respectively (Fig. 1G). This is in agreement with previous work suggesting that neurites initially contain microtubules of mixed polarity (Yau et al., 2016) and demonstrates that developing neurites have less minus-end-out microtubules overall than the ∼50% reported for mature dendrites (Tas et al., 2017). Furthermore, the observed segregation indicates that the mechanisms that establish spatial segregation by orientation, likely involving MAPs or motors with a preference for crosslinking parallel microtubules, must already be operational early on in development (see Discussion).

## Some late stage 2 neurons have an axon-like neurite with a uniformly plus-end-out microtubule array

In addition, we commonly observed neurites (one or two per cell) with almost exclusively plus-end-out microtubules [Fig. 1E and F (left); Fig. S1C–E]. Although this is expected for axons (of stage 3 neurons), it was interestingly also the case for some stage 2b neurons, in which we could find neurites with predominantly or exclusively plus-end-out microtubules. These neurites of uniform polarity were not necessarily the longest neurite in stage 2b neurons, and this correlation emerged more prominently only later in development (Fig. S2A). Importantly, we cannot determine whether the axon-like neurites in stage 2b neurons actually go on to become the axon or whether this is a transient state, as reported previously (Burute et al., 2022; Jacobson et al., 2006). Furthermore, the appearance of these axon-like neurites might explain why stage 2b and stage 3 neurons had similar levels of minus-end-out microtubules (median values of 0.28 and 0.29; Fig. 1G). As reported previously (Yau et al., 2016), axons of stage 3 neurons still contained a few remaining minus-end-out microtubules. Sometimes these axons and axon-like neurites had a small bundle of minus-end-out microtubules proximal to the soma (Fig. S2B,C). This suggests that plus-end-out uniformity emerges distally first in these neurites, perhaps by retrograde sliding of these minus-end-out microtubules (see Discussion). In light of these axon-like neurites, we also compared the fraction of minus-end-out microtubules per neurite while excluding axons and axon-like neurites (defined as any neurite with less than 10% minus-end-out localizations), which resulted in overall similar distributions, but slightly higher median values for later developmental stages (0.30 and 0.33 for stage 2b and stage 3, respectively; Fig. S2D).

## Stable and labile microtubules are segregated from early on in development

In the dendrites of mature neurons, there is a relationship between the stability of a microtubule and its orientation, with stable microtubules preferentially oriented minus-end-out (Tas et al., 2017). To determine at which stage this organization emerges, we fixed neurons at similar time points after plating to those described above, stained for tyrosinated tubulin (a marker for labile microtubules) and acetylated tubulin (a marker for stable microtubules), and performed ultrastructure expansion microscopy (U-ExM; Gambarotto et al., 2019). This revealed that, much like microtubules of opposite orientation (Fig. 1), microtubules decorated by different PTMs also segregated into different bundles in stage 2 and stage 3 (minor) neurites (Fig. 2). To quantify this segregation, we exploited the improved *z*-resolution of expansion microscopy to construct radial distribution maps of the intensities of these two PTMs and averaged these across (minor) neurites (Katrukha et al., 2021). This revealed that acetylated and tyrosinated microtubules indeed have different radial distributions, with acetylated microtubules being, on average, enriched centrally compared to tyrosinated microtubules, particularly later in development (Fig. 2D,F). In stage 2a cells, the segregation was present but less evident in the averaged profile. This is likely because, although we observed the two subsets to be segregated at this stage, we did not always see the central–peripheral distinction in these early neurites (Fig. 2A,B), similar to what we observed with the segregation of microtubules of opposite orientation (Fig. 1H). Thus, the bundles of acetylated (stable) microtubules might indeed correspond to the bundles of minus-end-out microtubules, whereas the bundles of tyrosinated (labile) microtubules might correspond to bundles of plus-end-out microtubules.

Journal of Cell Science

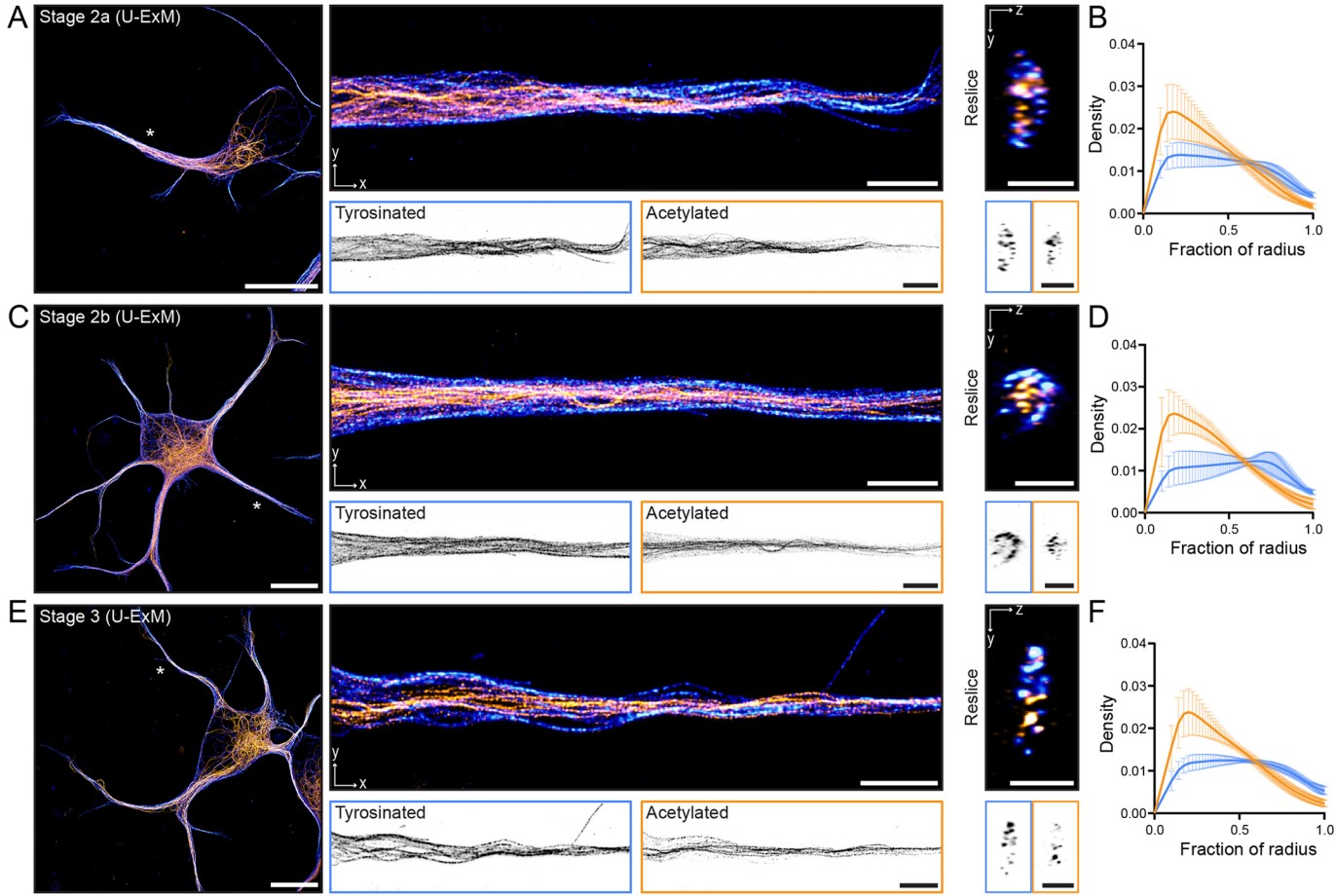

**Fig. 2. Stable and labile microtubules are segregated from early on in neuronal development.** (A) Left: U-ExM image of a stage 2a cell showing both acetylated (orange) and tyrosinated (blue) microtubules. Scale bars: 10 μm (corrected for expansion). Asterisk indicates neurite shown to right. Middle: U-ExM image of a single neurite from this cell. Single-channel images are shown below. Scale bars: 2 μm (corrected for expansion). Right: *y-z* cross-section of the neurite shown. Scale bars: 1 μm (corrected for expansion). (B) Radial distribution of the intensities of acetylated (orange) and tyrosinated (blue) microtubules averaged across *n*=9 neurites from *N*=6 stage 2a neurons. (C) Left: U-ExM image of a stage 2b cell showing both acetylated (orange) and tyrosinated (blue) microtubules. Asterisk indicates neurite shown to right. Middle: U-ExM image of a single neurite from this cell. Single-channel images are shown below. Right: *y-z* cross-section of the neurite shown. Scale bars as in A. (D) Radial distribution of the intensities of acetylated (orange) and tyrosinated (blue) microtubules averaged across *n*=11 neurites from *N*=5 stage 2b neurons. (E) Left: U-ExM image of a stage 3 cell showing both acetylated (orange) and tyrosinated (blue) microtubules. Asterisk indicates neurite shown to right. Middle: U-ExM image of a single neurite from this cell. Single-channel images are shown below. Right: *y-z* cross-section of the neurite shown. Scale bars as in A. (F) Radial distribution of the intensities of acetylated (orange) and tyrosinated (blue) microtubules averaged across *n*=8 neurites from *N*=5 stage 3 neurons. Data in B,D,F are presented as mean±s.d.

To directly query the orientation of stable microtubules, we selectively removed labile microtubules using nocodazole before performing motor-PAINT (Tas et al., 2017). We first verified the efficacy and non-lethality of the nocodazole treatment on neurons early on in development and saw that we could indeed selectively depolymerize the tyrosinated (labile) microtubules, while retaining the acetylated (stable) microtubules (Fig. 3A). Using motor-PAINT, we observed a higher fraction of minus-end-out microtubules in non-axon(-like) neurites after nocodazole treatment as compared to the control in stage 2b (median values of 0.30 and 0.48 without and with nocodazole, respectively) and stage 3 (median values of 0.33 and 0.40 without and with nocodazole, respectively) neurons (Fig. 3D–G; Figs S2D, S3A,B). We also saw similar values for the fraction of stable microtubules in non-axon neurites (i.e. dendrites) of stage 4 neurons after nocodazole treatment (medians of 0.32 and 0.45 without and with nocodazole, respectively; Fig. S3C–E). This indicates that the stable microtubules in these neurites are indeed preferentially oriented minus-end-out compared to the total population of microtubules, although there are still more

plus-end-out stable microtubules during these stages. Given that ∼70% of the total population of microtubules is plus-end-out and that stable microtubules are less plus-end-out (∼50–60%), our findings suggest that dynamic microtubules are mostly plus-end-out. Indeed, a previous study has shown that EB protein comets, which are found at the growing plus-ends of dynamic microtubules and thus predominantly indicate the orientation of these microtubules (Jansen et al., 2023; Yau et al., 2016), are mostly anterograde (i.e. the microtubules are plus-end-out) (Yau et al., 2016).

### Stable microtubules are oriented plus-end-out in early stage 2 neurites and often connected to the centrosome

Next, we repeated this procedure in stage 2a neurons. It was apparent that there were fewer stable microtubules in these neurons, and we observed shorter tracks in motor-PAINT after nocodazole treatment (Fig. 3B; Fig. S3A,B), which could imply that the microtubules are stabilized in shorter stretches at this stage. Interestingly, unlike for stage 2b and stage 3 neurons, we

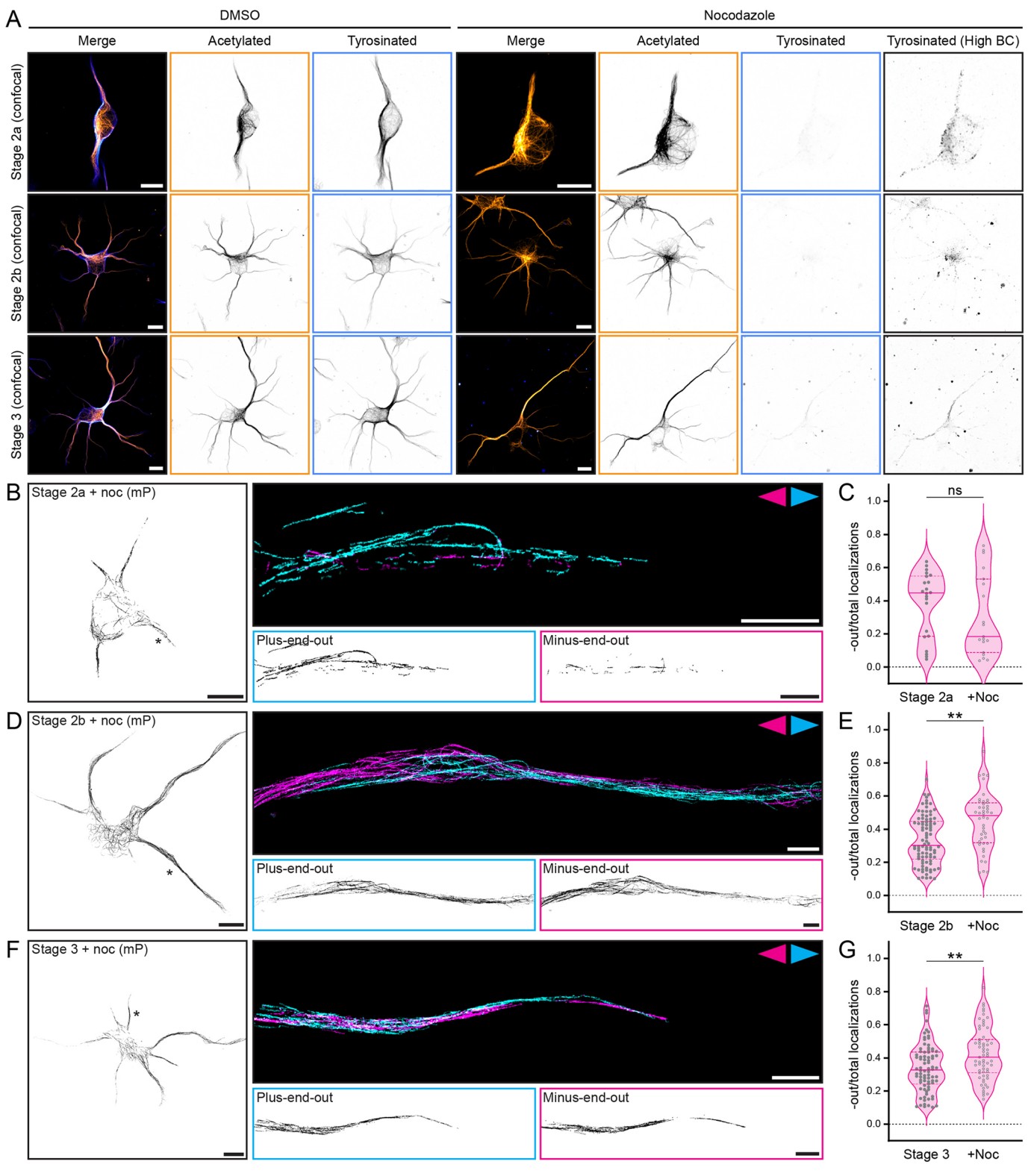

**Fig. 3.** See next page for legend.

observed a lower fraction of minus-end-out microtubules in stage 2a neurons after nocodazole treatment as compared to control (median values of 0.45 and 0.18 without and with nocodazole, respectively; Fig. 3C), although this difference was not statistically significant, perhaps because of the smaller sample size (fewer neurites per cell). Moreover, four of the six neurites with 50% or more minus-end-out microtubules after nocodazole treatment belong to the same cell, making this cell somewhat of an outlier. These results suggest that stable microtubules are initially predominantly oriented plus-end-out. Perhaps stable microtubules are originally nucleated at the centrosome (or microtubules nucleated at the centrosome are stabilized while anchored) such that they are plus-end-out and then

**Fig. 3. Stable microtubules are more minus-end-out than total microtubules in the minor neurites of stage 2b and stage 3 neurons, but not stage 2a neurons.** (A) Nocodazole treatment is efficient and non-lethal in developing neurons. Example confocal images showing tyrosinated (blue) and acetylated (orange) microtubules in stage 2a, 2b and 3 neurons treated with 4 µM nocodazole for 2.5 h or an equivalent amount of DMSO, showing the reduction in intensity of tyrosinated microtubules after nocodazole treatment. Imaging conditions were kept the same in all conditions, and the brightness and contrast were scaled the same except for the 'High BC' column, for which the brightness and contrast were greatly increased to show any remaining tyrosinated microtubule signal. Scale bars: 10 µm. (B) Left: total tracks after filtering (all orientations) of a stage 2a neuron imaged using motor-PAINT (mP) after nocodazole (noc) treatment. Scale bar: 10 µm. Asterisk indicates neurite shown to right. Right: a motor-PAINT reconstruction of a neurite from this cell with microtubules pseudo-coloured based on whether their plus-end is oriented towards (magenta) or away from (cyan) the soma. Single-channel images are shown below. Scale bars: 2 µm. (C) Quantification of the fraction of localizations constituting minus-end-out tracks over the total amount of localizations for stage 2a neurons without and with (+Noc) nocodazole treatment. Each dot represents one neurite. Medians (solid line; control 0.45, +Noc 0.18) and interquartile ranges (dashed lines; control 0.19–0.55, +Noc 0.09–0.53) are shown. $n=22$, 19 neurites from $N=9$, 8 cells for control and nocodazole-treated cells, respectively. Groups compared by two-tailed Mann–Whitney test (ns, not significant). (D) Left: total tracks after filtering (all orientations) of a stage 2b neuron imaged using motor-PAINT after nocodazole treatment. Scale bar: 10 µm. Asterisk indicates neurite shown to right. Right: a motor-PAINT reconstruction of a neurite from this cell with microtubules pseudo-coloured based on whether their plus-end is oriented towards (magenta) or away from (cyan) the soma. Single-channel images are shown below. Scale bars: 2 µm. (E) Quantification of the fraction of localizations constituting minus-end-out tracks over the total amount of localizations for non-axon-like neurites in stage 2b neurons without and with (+Noc) nocodazole treatment. Each dot represents one neurite. Medians (solid line; control 0.30, +Noc 0.48) and interquartile ranges (dashed lines; control 0.22–0.49, +Noc 0.32–0.56) are shown. $n=90$, 44 neurites from $N=18$, 11 cells for control and nocodazole-treated cells, respectively. Groups compared by two-tailed Mann–Whitney test (**$P≤0.01$). (F) Left: total tracks after filtering (all orientations) of a stage 3 neuron imaged using motor-PAINT after nocodazole treatment. Scale bar: 10 µm. Asterisk indicates neurite shown to right. Right: a motor-PAINT reconstruction of a neurite from this cell with microtubules pseudo-coloured based on whether their plus-end is oriented towards (magenta) or away from (cyan) the soma. Single-channel images are shown below. Scale bars: 2 µm. (G) Quantification of the fraction of localizations constituting minus-end-out tracks over the total amount of localizations for non-axon neurites in stage 3 neurons without and with (+Noc) nocodazole treatment. Each dot represents one neurite. Medians (solid line; control 0.33, +Noc 0.40) and interquartile ranges (dashed lines; control 0.24–0.44, +Noc 0.31–0.56) are shown. $n=82$, 63 neurites from $N=17$, 14 cells for control and nocodazole-treated cells, respectively. Groups compared by two-tailed Mann–Whitney test (**$P<0.01$).

later released, for example via severing, to allow them to reverse their orientation to minus-end-out (e.g. via sliding) in dendrites. Indeed, this has been proposed previously for microtubules in both axons and dendrites (Ahmad et al., 1999; Ahmad and Baas, 1995; Baas and Yu, 1996; Sharp et al., 1995), and it is known that the centrosome is initially active in stage 1 neurons and then inactivated during neuronal development preceding axon outgrowth (i.e. stage 3) (Stiess et al., 2010). Alternatively, the plus-end-out stable microtubules could be selectively depolymerized to make way for newly nucleated minus-end-out microtubules, or they could be simply outnumbered by these minus-end-out microtubules.

To study whether stable microtubules are initially anchored at the centrioles, we fixed neurons at similar time points after plating, stained for acetylated tubulin and again used U-ExM (Gambarotto et al., 2019) to study the microtubule network around the centrioles. Expansion microscopy is well-suited for this because it affords

improved *z*-resolution, which is essential to resolve microtubules in the dense network surrounding the centrioles. This revealed that, early on in development (stage 1 and stage 2a), long acetylated microtubules were emanating from the centrioles (Fig. 4; Movies 1, 2). Later in development, the amount of long acetylated microtubules apparently anchored at the centrioles decreased until predominantly short stumps of acetylated microtubules (less than 0.5 µm when corrected for expansion factor) were observed around the centrioles in stage 2b and stage 3 neurons (Fig. 4; Movies 3, 4). Quantifying the absolute length distribution of the anchored microtubules is challenging, but we observed this phenomenon in at least six cells from three rats for each stage. In addition, given that we specifically stained for acetylated tubulin (a marker for stable microtubules), it is possible that other non-acetylated and thus perhaps dynamic microtubules are also associated with the centrioles. It was also apparent that, while in stage 1 the microtubules are anchored at both the mother and the daughter centriole, from stage 2a onwards, the microtubules or their remnants were preferentially anchored at one of the two centrioles. The stumps observed around centrioles in stage 2b and stage 3 neurons could be remnants of severed or released microtubules, suggesting that stable microtubules might be reoriented by sliding after release from the centrosome rather than by depolymerization and repolymerization.

## Stable microtubules undergo extensive sliding and polarity reversal during neuronal development

To better examine how stable microtubules are reorganized during the transition from stage 2a to stage 2b, we electroporated freshly dissociated neurons with stable microtubule-associated rigor kinesin (StableMARK; Jansen et al., 2023), a mutant non-motile kinesin-1 motor that specifically binds to stable microtubules, before plating and then imaged between 4 h and 27 h after plating using total internal reflection fluorescence (TIRF) microscopy (Fig. 5A,B). We observed extensive retrograde flow (Fig. 5C–E; Movie 5) in stage 2b neurons, as has also been observed for the total population of microtubules (Burute et al., 2022; Schelski and Bradke, 2022). The median speed of this flow was 0.15 µm/min (interquartile range of 0.10–0.24 µm/min; Fig. 5E), in agreement with previously reported values (Burute et al., 2022; Schelski and Bradke, 2022). Although StableMARK can induce microtubule bundling at high concentrations (Jansen et al., 2023), we chose cells with the lowest possible expression levels of StableMARK. Despite this, there was some apparent bundling in the neurons, which we believe to be non-aberrant given that it resembled that of acetylated microtubules in non-expanded cells (Fig. 2).

In addition, we observed highly motile stable microtubules that exhibited extensive sliding in the soma and neurites, particularly in stage 2a and stage 2b neurons (Fig. S4; Movies 6, 7). There were also instances in which a bundle of stable microtubules from one neurite appeared to be directly linked to a focal point (likely the centrosome) and instances in which the ends of some of these microtubules appeared to detach from this point (Fig. S4; Movies 7, 8), in agreement with our expansion data (Fig. 4). Importantly, we observed many instances of microtubule polarity reversal, particularly in stage 2b neurons. For example, we could observe a stable microtubule exiting from one neurite and being slid towards another neurite, which it then entered with the opposite orientation (Fig. 5F; Movie 9), and we could observe a stable microtubule curling back on itself within a given neurite, which also reversed its orientation (Fig. 5G; Movies 7, 9). These observations thus demonstrate two means by which stable microtubules can reverse their orientation in late stage 2 neurons. Importantly, we could follow the same microtubule for several hours, arguing against the depolymerization

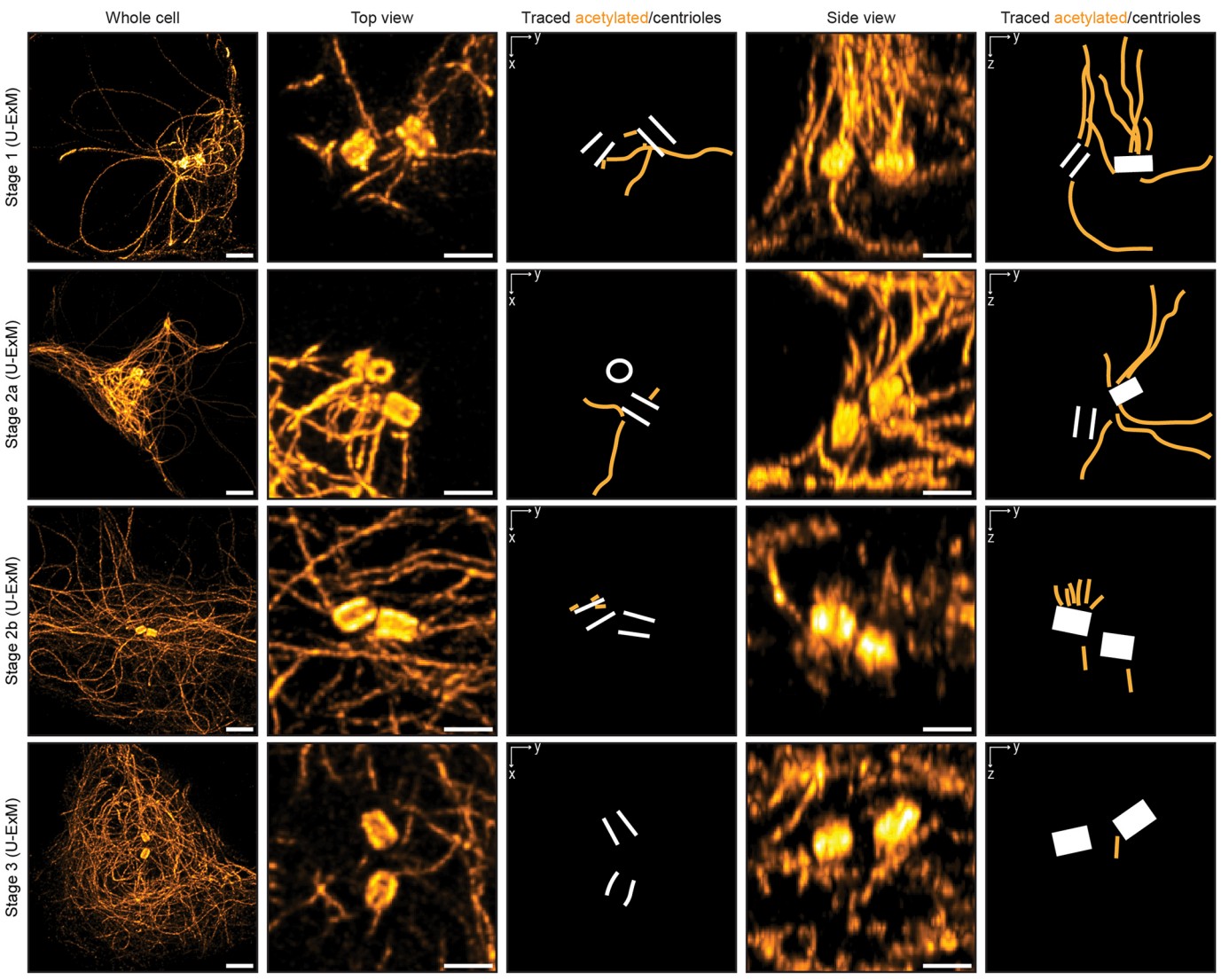

**Fig. 4. Stable microtubules are centrosomally anchored in stage 2a neurons but not later in development.** Maximum-projection images of acetylated microtubules in stage 1, stage 2a, stage 2b and stage 3 neurons (from top to bottom) acquired using U-ExM. Larger overviews of the soma are shown on the left. Maximum projections showing different views (top view, *x-y*; side view, *y-z*) of acetylated microtubules around the centrioles are shown in the subsequent columns. Representations in which the acetylated microtubules directly connected to the centrosome (centrioles marked in white) have been traced are also shown. Scale bars: 1 µm (corrected for expansion) in larger overviews, 0.5 µm (corrected for expansion) in top and side view zooms. Images shown are representative of six cells from three rats per stage.

and repolymerization of stable microtubules to reverse their orientation. We independently verified this long lifetime of a sub-population of microtubules using photoactivatable tubulin. This approach has previously been used to observe the motile behaviour of microtubules over time (Burute et al., 2022; Lu et al., 2016). Photoactivated tubulin not incorporated into microtubules is expected to quickly dissipate, whereas tubulin incorporated into persistent microtubules provides a persistent signal. We could indeed confirm the continued presence of signal after two or more hours, even as the microtubules that had incorporated this tubulin moved around in the cell (Fig. S5A,B; Movie 10).

Multi-day application of low doses (3–10 nM) of Taxol in developing neurons is known to induce the formation of multiple axons (Witte et al., 2008). Indeed, motor-PAINT experiments following such treatment revealed that neurons form an axon earlier on and tend to have more than one neurite with only plus-end-out microtubules (Fig. S2E,F). Given our observation that many stable microtubules start plus-end-out and then reorient to become minus-end-out in the

dendrites, we hypothesized that Taxol prevents this reorientation and thereby triggers the formation of multiple axons in which stable microtubules remain plus-end-out. To test this, we treated StableMARK-transfected neurons with the same low dose of Taxol and imaged them live using TIRF microscopy. Importantly, because Taxol also stabilizes microtubules and alters their lattice structure, ultimately leading to the redistribution of StableMARK (Jansen et al., 2023), we did not perform long-term incubations but instead observed the behaviour of microtubules during and immediately after Taxol treatment. This revealed that Taxol indeed dampens the sliding of stable microtubules (Fig. S5C; Movies 11, 12), suggesting that active reorientation of stable microtubules is required for dendrite formation.

# DISCUSSION
In this work, we mapped out the reorganization of the microtubule cytoskeleton that occurs during neuronal polarization (Fig. 5H). Emerging neurites of early stage 2 neurons already contain microtubules of both orientations, and these are typically

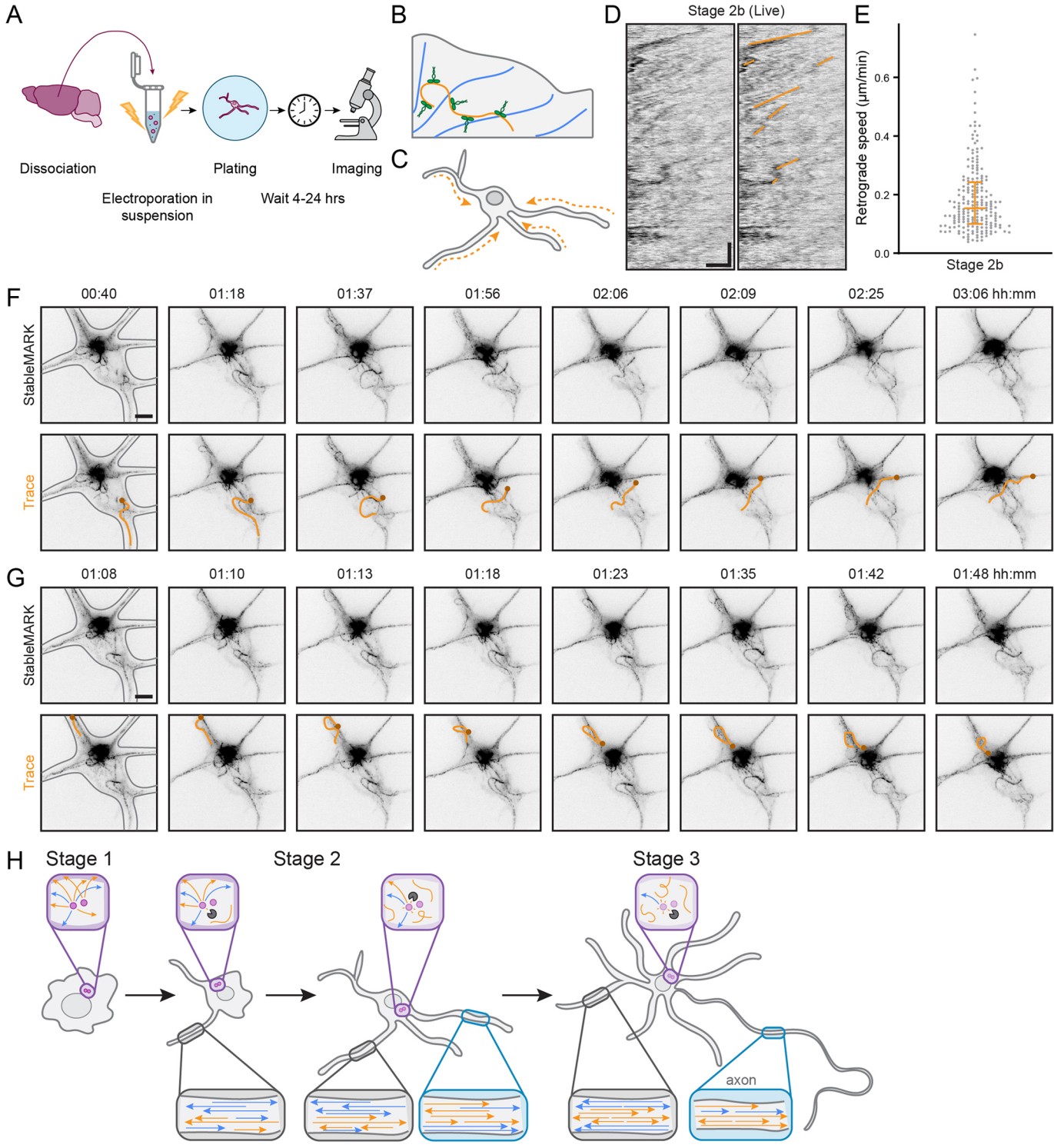

**Fig. 5.** See next page for legend.

segregated into bundles of locally uniform orientation. These emerging neurites also contain segregated networks of acetylated (stable) and tyrosinated (labile) microtubules. In later stage 2, stage 3 and stage 4 neurons, stable (nocodazole-resistant) microtubules are oriented more minus-end-out compared to the total (untreated) population of microtubules; however, in early stage 2 neurons, stable microtubules are preferentially oriented plus-end-out, likely because their minus-ends are still anchored at the centrioles at this stage. The

fraction of anchored stable microtubules decreases during development, while the appearance of short stumps of microtubules attached to the centrioles suggests that these microtubules might be released by severing. Finally, using live-cell imaging we observed the retrograde flow of stable microtubules and two mechanisms by which stable microtubules reverse their orientation. In addition to these phenomena, which predominantly describe the development of the dendrites, we also identified axon-like neurites with (almost)

**Fig. 5. Stable microtubules undergo retrograde flow and reverse their orientation during the early stages of neuronal development.**
(A) Schematic showing how dissociated neurons from embryonic rat hippocampi were electroporated with StableMARK prior to plating on laminin-coated coverslips and were imaged shortly thereafter. (B) Schematic showing StableMARK (green) bound to stable microtubules (orange), whereas labile microtubules (blue) are undecorated. (C) Schematic showing the retrograde flow (arrows) of stable microtubules observed in young neurons. (D) Kymographs of this retrograde flow of stable microtubules (labelled using StableMARK, inverted greyscale) in a stage 2b neuron (left) with examples of the lines used to determine the speed of retrograde flow overlaid in orange (right). Scale bars: 2 µm (horizontal) and 30 min (vertical). Corresponds to Movie 5. (E) Quantification of the speed of retrograde flow from $n$=30 neurites in $N$=7 stage 2b neurons. Each dot represents one line as traced in D. The median (0.15 µm/min) and interquartile range (0.10–0.24 µm/min) are shown. (F) An example of a stable microtubule in a stage 2b neuron reversing its orientation by exiting from one neurite and entering another. The microtubule of interest is traced in orange in the bottom row, with the same end marked by a small circle in each frame to indicate orientation. First frame also has the cell outline in grey. Times shown above each frame are in hours:minutes. Scale bar: 5 µm. Corresponds to Movie 9. (G) An example of another stable microtubule in the same neuron reversing its orientation by turning back on itself within one neurite. The microtubule of interest is traced in orange in the bottom row, with the same end marked by a small circle in each frame to indicate orientation. First frame also has the cell outline in grey. Times shown above each frame are in hours:minutes. Scale bar: 5 µm. Corresponds to Movie 9. (H) Stable microtubules reverse their orientation during neuronal development by dissociating from the centrosome. Schematic showing the stages of neuronal development and the accompanying changes in the microtubule cytoskeleton. It is now apparent that microtubules of opposite orientation (indicated by arrowheads) are segregated from early on in development, but that stable microtubules (orange) are initially mostly plus-end-out in stage 2a and later reverse their orientation to become minus-end-out. This is facilitated by their detachment from the centrosome (purple), which allows them to be slid into neurites with their minus-ends leading. Labile microtubules (blue) remain largely plus-end-out throughout development.

uniformly plus-end-out microtubules in late stage 2 neurons, prior to axon specification.

Our results for microtubule orientations in developing neurites are consistent with earlier work that used EB comet tracking after laser severing in stage 2 and 3 neurons. This work revealed that the fraction of comets moving towards the soma increases after laser severing, which indirectly suggests that minus-end-out microtubules are often more stable (Yau et al., 2016). Our direct analysis of the orientation of stable microtubules revealed very similar fractions of stable minus-end-out microtubules in stage 2b and stage 3 neurons, but these values are lower than what has been reported for mature neurons (Tas et al., 2017). This suggests that the population of stable minus-end-out microtubules continues to increase during development.

Previous work reported microtubules emanating from the centrioles of developing neurons (Stiess et al., 2010; Yu et al., 1993) but could not selectively detect stable microtubules. Our work now shows that in stage 1 and somewhat in early stage 2, acetylated (stable) microtubules are also anchored at the centrioles. This explains why stable microtubules are predominantly oriented plus-end-out at these stages. During stage 2, many stable microtubules begin to reverse their orientation and are no longer anchored at the centrioles. Instead, we found an increasing density of short acetylated microtubules attached to the centrioles during this stage. Together with the very long lifetime of stable microtubules observed with live-cell imaging, this suggests that stable microtubules are released from the centrosome via severing by spastin, katanin or fidgetin, followed by redistribution and reorientation. We favour severing over the recently described CAMSAP-mediated release of microtubules from the γ-tubulin ring

complex (γTuRC; Rai et al., 2024) because of the presence of these microtubule stumps. It is unlikely that CAMSAPs, which are microtubule minus-end-binding proteins, would leave such a microtubule remnant at the centrioles. Taken together, our results show that the initial population of stable microtubules follows the centrosomal-release-and-reorientation model proposed by Baas and colleagues (Baas and Yu, 1996; Yu et al., 1993). In the future, it will be interesting to investigate what triggers the activity of these severing enzymes, as well as the potential effects on microtubule polarity during development in cells depleted of them. It will also be exciting to study whether these stable microtubules are nucleated directly as stable microtubules and perhaps acetylated more easily due to their lattice conformation or if they are stabilized and modified after some period of growth.

Once released, the plus-end-out stable microtubules must reverse their orientation, whereas the dynamic/labile microtubules do not. We provide evidence that stable microtubules slide between or within neurites to reverse their orientation rather than depolymerize and repolymerize. What remains unclear, however, is how this sliding is effectuated. In the example shown (Movie 9), the microtubule end covers a distance of ∼15 µm in ∼4 h (∼60 nm/min on average). During this time, the microtubule end undergoes bursts of transient motility and periods in which it is rather immobile. We speculate that this motility is driven by microtubule-based motors, perhaps in combination with hitchhiking on a growing plus-end of a dynamic microtubule (Mattie et al., 2010). Multiple motor proteins, which move at a range of speeds, have been implicated in microtubule polarity sorting and sliding during neuronal development (del Castillo et al., 2015; He et al., 2020; Kahn et al., 2015; Lin et al., 2012; Liu et al., 2010; Lu et al., 2013; Mattie et al., 2010; Muralidharan and Baas, 2019; Schelski and Bradke, 2022; Weiner et al., 2020). One of these is kinesin-1, a plus-end-directed motor that preferentially binds to and moves on stable microtubules (Cai et al., 2009; Reed et al., 2006) and that has been shown to slide microtubules in multiple cell types, including neurons (He et al., 2020; Jolly et al., 2010; Lu et al., 2013; Winding et al., 2016). This motor could crosslink microtubules or crosslink a microtubule with, for example, the actin cortex, and, by walking on the stable microtubule towards its plus-end, push this microtubule with its minus-end leading into a new neurite (Kapitein and Hoogenraad, 2015). Given the preferentially minus-end-out orientation of stable microtubules in mature dendrites, kinesin-1 could be a good candidate to effectuate the necessary sliding, although its unloaded speed is much higher than those of the movements we observed. Alternatively, a motor that moves along labile microtubules but that associates with CAMSAPs at the minus-end of a stable microtubule could help drag this minus-end into a minor neurite. There is indeed evidence that at least the minus-end-directed kinesin-14 can interact with CAMSAP2 (Cao et al., 2020). It is likely that a combination of motors orchestrates the sliding of stable microtubules between neurites and within neurites to facilitate their microtubule polarity reversal. To solve this puzzle, it might be helpful to image StableMARK during neuronal development upon treatment with inhibitors or activators of different motor proteins such as kinesore (activator of kinesin-1; Randall et al., 2017), monastrol (inhibitor of kinesin-5; Mayer et al., 1999), or dynarrestin and ciliobrevin D (inhibitors of dynein; Firestone et al., 2012; Höing et al., 2018).

Within neurites, we observed a clear spatial separation both between labile and stable microtubules and between plus-end-out and minus-end-out microtubules. To achieve this, it is likely important to crosslink microtubules of the same polarity. Interestingly, the microtubule bundler TRIM46 is one of the two MAPs known so

far to crosslink microtubules with a preferred relative orientation. It preferentially crosslinks parallel microtubules (Harterink et al., 2019; Van Beuningen et al., 2015), but oscillates between neurites in stage 2 before settling in the axon (Burute et al., 2022), making it an unlikely candidate for bundling parallel microtubules in the minor neurites and dendrites; however, there are other microtubule-binding TRIM family members that might fulfil this role (Glover et al., 2024; Short and Cox, 2006). Parallel bundling by TRIM46 has also been shown to induce stabilization of the bundled microtubules (Van Beuningen et al., 2015). Such bundling-induced stabilization might contribute to the amplification of the initial population of centriole-derived and polarity-reversed stable microtubules, as the number of stable microtubules continues to increase after the inactivation of the centrosome.

Finally, we observed the presence of axon-like neurites with (almost) uniformly plus-end-out microtubule arrays in late stage 2 neurons prior to axon specification. Importantly, these neurites were not always the longest of the cell, and from the present study, we cannot deduce whether these will indeed become the axon, but these observations are consistent with previous reports of transient polarization in this stage (Burute et al., 2022; Jacobson et al., 2006). It will thus be interesting to see whether this plus-end-out uniformity coincides with the waving of kinesin-1 and TRIM46, two axonal markers, that occurs in stage 2 neurons (Burute et al., 2022).

The present study was conducted in dissociated rat hippocampal neurons, an important model system used by many labs worldwide. Nevertheless, it is interesting to also consider how this process might be similar or different *in situ*. In the brain, there are many factors involved in directing neuronal migration, differentiation and axon specification. For example, positive and negative guidance cues help direct axon growth, and stiffness gradients are also known to play a role both in neuronal differentiation (Engler et al., 2006) and axon specification (Burute et al., 2022). As such, we expect there to be less variability between neurites and perhaps a more efficient axon specification step such that stage 2 is rather brief. However, given that neurons arise by cell division from neuronal precursor cells, we expect their centrosomes to start as active microtubule-nucleating centres and progressively inactivate, much like the dissociated hippocampal neurons we used. This means it is not implausible that stable microtubules are similarly originally nucleated from and/or anchored at the centrosome, making them plus-end-out, and then later released to allow them to reverse their orientation. However, more tools are needed to better examine the organization and orientation of microtubules in tissue to validate this. At present, this study provides the most in-depth examination of microtubule reorganization during development to date and exploits the relatively easy manipulation and high-resolution imaging of cultured rat hippocampal neurons to gain insights into how stable microtubules are made and reoriented during the course of development.

## MATERIALS AND METHODS

### Cell culture

All experiments were approved by the DEC Dutch Animal Experiments Committee (Dier Experimenten Commissie), performed in line with institutional guidelines of Utrecht University, and conducted in agreement with Dutch law (Wet op de Dierproeven, 1996) and European regulations (Directive 2010/63/EU). For neuron cultures, 18 mm coverslips (no. 1.5 high precision) were cleaned and coated with poly-L-lysine (37.5 µg/ml; Sigma-Aldrich, P8920) and laminin (1.25 µg/ml; Roche, 11243217001). Subsequently, hippocampi were dissected from embryonic day 18 rat brains (sex not determined) and dissociated. For all experiments without

electroporation, the resulting primary hippocampal neurons were then plated at a density of 50,000 cells/well (12-well plate) onto the treated coverslips in neurobasal medium (NB; Gibco, 21103049) supplemented with 2% B27 (Gibco, 17504001), 0.5 mM glutamine (Gibco, 25030-081), 15.6 µM glutamate (Sigma-Aldrich, G1251) and 1% penicillin-streptomycin (Gibco, 15140-122) (full medium). These cells were maintained at 37°C and 5% $CO_2$ until use. For more details, see Kapitein et al. (2010).

### Drug treatment

Nocodazole treatment was performed as described previously (Tas et al., 2017). Cells were treated with 4 µM nocodazole for 2.5 h at 37°C and 5% $CO_2$. For immunofluorescence assays, control samples were treated with an equivalent amount of DMSO (∼0.04%). For motor-PAINT assays, no DMSO was added to control samples. Taxol treatment for motor-PAINT experiments was performed as follows: cells were treated with 10 nM Taxol ∼6 h after plating and maintained in this Taxol-containing medium until the sample was prepared for motor-PAINT on the indicated day *in vitro* (DIV). For live-cell Taxol experiments on StableMARK-transfected neurons, 10 nM Taxol (or an equivalent amount of DMSO, ∼0.04%) was added acutely during imaging (after 30 min) by replacing a portion of the medium with Taxol-containing medium.

### Immunofluorescence

To verify the efficiency of our nocodazole treatment, neurons were fixed on DIV0, DIV1, DIV2 and DIV3 as follows. First, soluble tubulin was extracted by subjecting cells to an extraction step (60 s, buffer pre-warmed to 37°C) with 0.3% Triton X-100 and 0.1% glutaraldehyde in MRB80 (80 mM K-PIPES, 1 mM EGTA, 4 mM $MgCl_2$; pH 6.80). This buffer was then exchanged to allow for fixation (10 min, buffer pre-warmed to 37°C) using 4% formaldehyde and 4% w/v sucrose in MRB80. Cells were then rinsed with PBS, permeabilized using 0.2% Triton X-100 in PBS for 10 min, rinsed with PBS, quenched three times for 5 min in 5 mg/ml sodium borohydride in PBS, rinsed with PBS and blocked for 30 min in 3% w/v bovine serum albumin (BSA) in PBS. Samples were then incubated in primary antibodies (see Table S1) for 1.5–2 h at room temperature, washed with PBS and then incubated with secondary antibodies (see Table S1) for 1–1.5 h. Finally, samples were washed with PBS and then MilliQ water, dried and lastly mounted in ProLong Diamond Antifade mountant (Thermo Fisher Scientific P36970). The following antibodies were used in this study: rat anti-tyrosinated tubulin (Abcam, ab6160), rabbit anti-acetylated tubulin (Cell Signaling Technology, 5335), Alexa Fluor 647-conjugated goat anti-rat IgG (Life Technologies, A-21247), Alexa Fluor 488-conjugated goat anti-rabbit IgG (Thermo Fisher Scientific, A-11008), Alexa Fluor 594-conjugated goat anti-rat IgG (Thermo Fisher Scientific, A-11007). See Table S1 for the dilutions used for different application (confocal or U-ExM).

### Confocal imaging

Fixed samples were imaged on an LSM 700 laser scanning confocal microscope (Zeiss): an Axio Observer Z1 inverted microscope with a motorized stage, a plan-apochromat 63× oil immersion objective (NA 1.40) and 488/555/639 nm laser lines. The following filters were used: bandpass (BP) 490–555 nm, BP 592–622 nm and long pass (LP) 640 nm. Detection was performed using a multialkali photomultiplier tube. Components were controlled using ZEN 2012 (Zeiss).

### Electroporation

Electroporation was performed immediately after dissociation, prior to plating. Dissociated neurons were pelleted (200 *g*, 5 min) and then gently resuspended in Nucleofector solution (Amaxa Biosystems/Lonza Bioscience) freshly supplemented with 20% fetal bovine serum (Corning, 35_079_CV). The neurons were then mixed with 0.5 µg StableMARK (Jansen et al., 2023; Addgene 174649) or with 0.3 µg PAGFP–α-tubulin (Kuijpers et al., 2016) and 0.3 µg MARCKS–RFP (Kuijpers et al., 2016) in an electroporation cuvette with 200,000–1,000,000 neurons per reaction. These neurons were then electroporated using the Lonza Nucleofector 2b on the O-003 (rat hippocampal neurons) setting. After adding full medium, these cells were plated on cleaned and coated 25 mm coverslips (no. 1.5 high precision; see above) at a density of 167,000 cells/well (6-well plate).

## Live-cell TIRF imaging

For live-cell TIRF imaging (with and without Taxol), neurons on 25 mm coverslips were mounted in imaging rings with conditioned medium and sealed using an additional coverslip to maintain $CO_2$ levels. Images were acquired on an inverted Nikon Eclipse Ti-E microscope equipped with a 100× Apo TIRF oil immersion objective (NA 1.49), a Perfect Focus System, an ASI motorized stage MS-2000-XY and an iLas2 system (Gataca Systems) for azimuthal TIRF/oblique illumination imaging. For imaging, a Stradus 488 nm (150 mW; Vortran) laser was used together with the ET-GFP filter set (49002; Chroma). Images were projected onto the chip of an Evolve Delta 512 EMCCD camera (Photometrics) with a 2.5× intermediate lens (Nikon C mount adapter 2.5×) at a magnification of 0.065 µm/pixel and 16-bit pixel depth. To keep cells at 37°C, we used a stage-top incubator (model STXG-PLAMX-SETZ21L; Tokai Hit). StableMARK images were acquired with 6.5% laser power and 200 ms exposure time with a 1 min interval. Components were controlled using MetaMorph 7.10 (Molecular Devices).

## Live-cell spinning disc imaging

For live-cell spinning disc imaging, neurons on 25 mm coverslips were mounted in imaging rings with conditioned medium and sealed using an additional coverslip to maintain $CO_2$ levels. Images were acquired on a Nikon Eclipse Ti2-E microscope equipped with a 100× Plan Apo λD oil immersion objective (NA 1.45), a Perfect Focus System, an ASI motorized stage MS-2000-XYZ with Piezo Top Plate, a Yokogawa confocal spinning disc unit CSU-W1-T1 and an Ilas FRAP module (Gataca Systems). To keep cells at 37°C, we used a stage-top incubator (model STXG-PLAMX-SETZ21L; Tokai Hit). Components were controlled using MetaMorph 7.8 (Molecular Devices). Using the FRAP module, four regions of interest (ROIs) of 1 µm thickness were drawn perpendicular to one or more neurites, creating a barcode pattern. All ROIs were illuminated simultaneously using a Vortran Stradus 405 (100 mW) laser at 20% laser power. Illumination settings were optimized to achieve the highest GFP signal in the ROI. After photoactivation, the cells were imaged for 3.5 h with a 1.5 min interval. Imaging was performed using Vortran Stradus 488 (150 mW) and Coherent OBIS 561 (150 mW) laser lines both at 5% laser power, ET5525/50 m (GFP) and ET630/75 m (mCherry) filters, and a Prime BSI sCMOS camera (Teledyne Photometrics).

## Protein purification

For motor-PAINT, DmKHC[1–421]–SNAP–6×His (Addgene 196975) was purified from *E. coli* BL21 cells as previously described (Deguchi et al., 2023). Briefly, after transformation, bacteria were cultured until $OD_{600} \approx 0.7$ at 37°C and 180 rpm. Cultures were then cooled to below 20°C, after which protein expression was induced with 0.15 mM IPTG at 18°C and 180 rpm overnight. Cells were then pelleted by centrifugation at 4500 $g$, snap frozen in liquid nitrogen and stored at −80°C until use. On the day of purification, cells were rapidly thawed at 37°C before being resuspended in chilled lysis buffer (50 mM sodium phosphate buffer supplemented with 5 mM $MgCl_2$, 5 mM imidazole, 10% v/v glycerol, 300 mM NaCl, 0.5 mM ATP and 1× EDTA-free cOmplete protease inhibitor; pH 8.0). Bacteria were lysed by sonication (five rounds of 30 s), supplemented with 2 mg/ml lysozyme (Sigma, 62971-10G-F), and then incubated on ice for 45 min. The lysate was clarified by centrifuging at 26,000 $g$ for 30 min before being incubated with equilibrated cOmplete His-tag purification resin for 2 h. Beads were then pelleted and resuspended in five column volumes of wash buffer (50 mM sodium phosphate buffer supplemented with 5 mM $MgCl_2$, 5 mM imidazole, 10% v/v glycerol, 300 mM NaCl, and 0.5 mM ATP; pH 8.0) four times. Finally, the resin was transferred to a gravity flow column (Thermo Fisher Scientific, 89898). Once settled, the wash buffer was allowed to elute before adding three column volumes of elution buffer (50 mM sodium phosphate buffer supplemented with 5 mM $MgCl_2$, 300 mM imidazole, 10% glycerol, 300 mM NaCl and 0.5 mM ATP; pH 8.0) to elute the protein. The eluent was collected, concentrated by spinning through a 3000 Da molecular-weight-cutoff (MWCO) filter and supplemented with 1 mM dithiothreitol (DTT) and 10% w/v sucrose. These aliquots were then flash frozen in liquid nitrogen and stored at −80°C. For labelling, protein was thawed and incubated with an additional 1 mM DTT for 30 min before adding 50 µM JF646–SNAP-tag ligand (Janelia

Materials) and incubating with rotation overnight. Finally, protein was exchanged into wash buffer (low imidazole) supplemented with 2 mM DTT and 10% w/v sucrose by spinning through a 3000 Da MWCO filter. This also removes excess dye molecules, as these are not retained by the filter. Concentration was determined with a BSA standard gel. All steps from lysis onwards were performed at 4°C.

## Motor-PAINT sample preparation

For this work, the motor-PAINT protocol described previously (Chen et al., 2022; Deguchi et al., 2023; Tas et al., 2017) was somewhat modified. Neurons were permeabilized for 1 min in extraction buffer [BRB80 (80 mM K-PIPES, 1 mM $MgCl_2$, 1 mM EGTA; pH 6.80) supplemented with 1 M sucrose and 0.15% v/v Triton X-100] pre-warmed to 37°C. An equal volume of pre-warmed fixation buffer (BRB80 supplemented with 2% w/v paraformaldehyde and 0.08% glutaraldehyde) was added to this (i.e. final paraformaldehyde concentration of 1% w/v and final glutaraldehyde concentration of 0.04%), and the solutions were mixed by gently pipetting for 1 min. Subsequently, the extraction/fixation buffer mixture was removed, and the sample was washed four times with pre-warmed wash buffer (BRB80 supplemented with 1 µM Taxol) for 1 min each time. The sample was then incubated with wash buffer with added phalloidin 405 (165 nM or 1:400; Thermo Fisher Scientific, A12379) and 0.1 µm TetraSpeck beads ($10^8$ beads or 1:1000; Life Technologies, T7279; sonicated before use) and incubated at 37°C for 8–10 min. Next, this solution was removed, and the chamber was washed three times for 1 min each time with wash buffer. Before adding, an aliquot of DmKHC[1–421]–SNAP motors was thawed and spun in an Airfuge (Beckman Coulter) at 20 psi (A-100/30 rotor) for 5 min in a pre-chilled rotor to remove any aggregates, transferred to a clean tube and kept on ice until use. Finally, the wash buffer was exchanged for pre-warmed imaging buffer [BRB80 supplemented with 583 µg/ml catalase (Sigma, C40-500MG), 42 µg/ml glucose oxidase (Sigma, G2133-10KU), 1.7% w/v glucose, 1 mM DTT (Thermo Fisher Scientific, R0861) 2 mM methyl viologen (Sigma, 856177-1G), 2 mM ascorbic acid (Sigma, A92902-100G), 1 µM Taxol (Enzo Life Sciences, BML-T104-0005) and 5 mM ATP (Sigma, A2383-5G)] containing 0.5–3.0 nM kinesin motors (added in bulk). The coverslip was then mounted onto an indented microscope slide with imaging buffer and sealed using Twinsil two-component dental glue (Picodent). After a 5 min incubation at 37°C to allow for hardening of the dental glue, the sample was brought to the microscope for imaging.

## Motor-PAINT imaging

Imaging was performed immediately after sample preparation at room temperature (20–23°C) on a Nikon Ti-E inverted microscope equipped with a 100× Apo TIRF oil immersion objective (NA. 1.49), a Perfect Focus System 3 (Nikon), and custom optics allowing for a tunable angle of incidence to perform (pseudo-)TIRF microscopy. A Lighthub-6 laser combiner (Omicron) with a 638 nm laser (BrixX 500 mW multi-mode, Omicron) and a 405 nm laser (LuxX 60 mW, Omicron) was used for excitation. Emission light was separated from excitation light using a quad-band polychroic mirror (ZT405/488/561/640rpc, Chroma) and a quad-band emission filter (ZET405/488/561/640m, Chroma). Detection was done using a Hamamatsu Flash 4.0v2 sCMOS camera. Image stacks of motors were acquired with a 60 ms exposure time, 12–13% laser power and 15,000–20,000 images per field of view. Components were controlled using µManager (Edelstein et al., 2014).

## Identification of cell stage

Cells were classified as belonging to different stages based on their morphology as determined from the phalloidin staining. The number of neurites, their (relative) lengths, how branched they were, and the presence or absence of a remaining lamellipodium were used to assign neurons into the different stages. Stage 1 neurons had no neurites and a large lamellipodium. Stage 2a neurons most commonly had 1–4 neurites, all less than 20 µm in length with no branches, and sometimes had a remaining lamellipodium. Stage 2b neurons typically had more and slightly longer neurites (up to ~50 µm in length) with one or two branches, with no single neurite significantly longer than the rest. Stage 3 neurons had one (or sometimes two in our culture system) neurite(s) much longer than the rest that often had a few branches along it. Stage 4 neurons had extensive dendritic branching in addition to the presence of an axon.

## Motor-PAINT analysis

Preceding motor localization, raw movies were processed by median filtering to remove static and very rapidly moving (i.e. diffusive) particles (https://github.com/HohlbeinLab/FTM2). Subsequently, motors were detected and tracked using the Laplacian of Gaussian (LoG) detector and overlap tracker from TrackMate (Tinevez et al., 2017). The output data tables were converted into Results tables compatible with Detection of Molecules (DoM) for further processing.

For drift correction, two or three Tetraspeck beads were chosen per field of view and localized using Detection of Molecules (DoM) version 1.2.5 (https://github.com/UU-cellbiology/DoM_Utrecht). These localizations were then filtered and processed as follows. In the case of multiple bead localizations in the same frame, only the one with the highest signal-to-noise ratio was kept. Outliers (>3 standard deviations away from the mean of the localizations in adjacent frames) were removed and replaced by a copy of the preceding localization. Subsequently, jumps of >150 nm were detected, which represent abnormal drift. A rolling window was then applied between jumps to remove fluctuations. After correcting for jumps, the average of the first 1000 bead localizations was subtracted from all the localizations to create a drift correction table that was then applied to the motor localizations.

Motor tracks were then filtered. Tracks were only kept if they had ≥6 localizations, covered a distance >200 nm, had a speed of 100–1500 nm/s and a ratio of the net:total displacement of 0.1–0.9. Subsequently, these tracks were filtered for angle to remove pauses or reversals in orientation (i.e. due to erroneously linked spots). This was done by calculating displacement vectors from every localization $i$ to localization $i+3$. The angle between each pair of consecutive vectors was then calculated and stretches of ≥4 frames in which this angle was <50° were retained.

The resulting particle table was then split into four separate tables depending on whether the track had a positive or negative displacement in $x$ and $y$, and each orientation was assigned a lookup table: tracks moving to the top left were cyan, tracks to the top right were blue, tracks to the bottom right were magenta and tracks to the bottom left were yellow. Each of these particle tables were reconstructed separately using DoM and merged.

Alternatively, for the neurite-based colour coding, further analysis was performed. A lasso selector tool was written in Python to select all particle localizations in a given primary neurite or its branches. Each neurite or branch was then rotated to be approximately horizontal, and the localizations from all the tracks within this neurite or branch were treated as a whole, median filtered and subjected to spline fitting to produce a line used as the midline of that neurite or branch. After determining which end of this midline is closer to the soma, the localizations of each track were projected onto the midline, and the start and end point of the track were compared to determine whether the track was moving towards or away from the soma along the midline. Subsequently, tracks from the branches of a given neurite were grouped to report a single value per neurite (including all its branches). Minus-end-out fractions were calculated as the number of localizations in minus-end-out tracks over the total number of localizations in that neurite. Values from filopodia (very thin protrusions <10 μm in length) were excluded, as well as those from axons and axon-like neurites (>90% plus-end-out) where indicated. These data were then exported and plotted using GraphPad Prism version 10.1.1.

Codes are available at https://github.com/UU-cellbiology/neuron_motorPAINT.

## Ultrastructure expansion microscopy

Neurons were fixed differently depending on whether the centrioles or neurites were being imaged. For imaging neurites, soluble tubulin was extracted by incubating cultured neurons for 1 min with 0.3% v/v Triton X-100, 1 M sucrose and 0.1% w/v glutaraldehyde in MRB80 and then fixed for 8 min with 2% w/v paraformaldehyde and 1 M sucrose in MRB80. Cells were then quenched twice for 5 min with 100 mM sodium borohydride in PBS and washed with PBS prior to proceeding. For imaging of the centrioles, neurons were incubated with pre-warmed MRB80 for 1 min, followed by fixation in ice-cold methanol and 6 mM EGTA on ice for 12 min. Hereafter, neurons were rehydrated using four PBS washes (60 s, 2 min, 5 min, 5 min).

Anchoring was performed by incubating fixed neurons in 1.4% w/v formaldehyde and 3% w/v acrylamide (Sigma-Aldrich, A4058) in PBS for 3 h at 37°C. After several PBS washes, neurons were incubated with the U-ExM gelation solution [19% w/v sodium acrylate, 10% w/v acrylamide (Sigma-Aldrich, A4058), 0.1% w/v N,N′-methylenebisacrylamide (Sigma-Aldrich, M1533), 0.5% APS w/v (Sigma-Aldrich, 215589), 0.5% w/v TEMED (Bio-Rad, 1610800) and 1× PBS] for 2–3 min on ice and then polymerized for 1 h at 37°C (Gambarotto et al., 2019). The 38% w/v sodium acrylate stocks were prepared as previously described (Damstra et al., 2022). Briefly, acrylic acid (Sigma-Aldrich, 147230) was neutralized with 10 M sodium hydroxide to a final pH of 7.5–8 and stored at −20°C. Gels were denatured in the presence of 345 mM SDS, 225 mM Tris (pH 9) and 500 mM NaCl for 1.5 h at 95°C. After denaturation, gels were washed two times for at least 30 min in ultrapure water, followed by two washes in PBS for at least 30 min with shaking. The expansion factor used for image calibration was calculated by dividing the gel diameter after several MilliQ washes (until the gels did not expand any further) by the coverslip size. For post-expansion labelling, gels were then cut into pieces and incubated with primary antibodies (see Table S1) diluted 1:250 in PBS supplemented with 2% w/v BSA and 0.1% w/v Triton X-100 overnight at room temperature with shaking. Hereafter, gels were washed 4×15 min in PBS supplemented with 0.1% v/v Triton X-100. Gels were then incubated with secondary antibodies (see Table S1) diluted to 1:500 in PBS supplemented with 2% w/v BSA and 0.1% v/v Triton X-100 for 3 h at room temperature with shaking and finally washed 4×15 min in 0.1% v/v Triton X-100 in PBS. Finally, gels were re-expanded overnight with several MilliQ washes.

Gels were mounted on 25 mm, no. 1.5H poly-L-lysine-coated coverslips in an imaging ring and imaged on a Leica TCS SP8 STED 3X microscope using a 63×/1.2 NA water-immersion objective. A white laser, pulsed at 80 MHz, was used for excitation. Leica GaAsP HyD hybrid detectors were used for fluorescence detection. Using Las X software (Leica), stacks with a lateral pixel size of 70–75 nm and an axial spacing of 180 nm were acquired using the frame-sequential mode.

## Other analysis and image processing

Confocal, TIRF and motor-PAINT images were processed in FIJI (ImageJ) version 1.54f. In all instances, brightness and contrast were adjusted linearly. The background of the kymograph in Fig. 5D was subtracted using a rolling ball radius of 20 and no smoothing. Images of neurites were made from the whole field of view using Edit>Selection>Straighten separately for each channel. Line profiles were created using Analyze>Plot profile for each channel using a line width of 10 pixels (~200 nm) and exporting the resulting list of values for plotting in GraphPad Prism version 10.1.1. Note that for line plots, data were normalized separately for each channel using the minimum and maximum values in that dataset.

Expansion images were drift corrected and deconvolved with Huygens Professional (v17.04, Scientific Volume Imaging), using the classic maximum likelihood estimation (CMLE) algorithm. Images were iterated with CMLE over 8–10 iterations with a signal-to-noise ratio of 15. In all cases except for the whole-cell image of the stage 2a neuron shown in Fig. 4, maximum projections were made using ~3–5 slices around the middle of the neurites or ~6–9 slices around the centrioles. For the whole-cell image of the stage 2a neuron in Fig. 4, a sum projection was used instead.

To trace individual microtubules emanating from centrioles in expanded neurons, deconvolved expansion image z-stacks were cropped to a quadratic area with sides of 8 μm (1.78 μm when corrected for expansion factor) centred around the centrioles in ImageJ (v1.54f). Cropped stacks were loaded into BigTrace (v0.39, https://github.com/ekatrukha/BigTrace) to verify their attachment to the centrioles in three dimensions. Microtubules that were clearly attached to the centrioles in the three-dimensional data were traced manually in the two-dimensional maximum projections for easy visualization.

To analyse the radial distribution of acetylated and tyrosinated microtubules in expanded neurites, deconvolved image stacks were processed using custom scripts in ImageJ (v1.54f) and MATLAB (R2024b) as described in detail elsewhere (Katrukha et al., 2021). Briefly, on maximum intensity projections (xy plane), we drew polylines of sufficient thickness (300 pixels) to segment out neurite portions 44 μm (10 μm when corrected for expansion factor) in length proximal to the cell soma. Using Selection>Straighten on the corresponding z-stacks generated straightened B-spline interpolated stacks of

the neurite sections. These *z*-stacks were then resliced perpendicularly to the neurite axis (*yz*-plane) to visualize the neurite cross-section. From this, we could semi-automatically find the boundary of the neurite in each slice using first a bounding rectangle that encompasses the neurite (per slice) and then a smooth closed spline (approximately oval). To build a radial intensity distribution from neurite border to centre, closed spline contours were then shrunken pixel by pixel in each *yz*-slice while measuring ROI area and integrated fluorescence intensity. From this, we could ascertain the average fluorescence intensity per contour iteration, allowing us to calculate a radial intensity distribution by calculating the radius corresponding to each area (assuming the neurite cross-section is circular). Note that the curves all start at 0 because no microtubules fit into a circle of radius 0 and very few microtubules fit into circles with the smallest radii.

## Statistical testing

*P* values are indicated in the appropriate figures and figure legends: ns, not significant; *$P \leq 0.05$; **$P \leq 0.01$; ***$P \leq 0.001$; ****$P \leq 0.0001$. Given the non-normality of the distributions, groups were compared using the Mann–Whitney test. The number of neurites and cells analysed is indicated in each figure legend. All quantified data except for Fig. S2F were obtained using neurons from at least three different rats.

### Competing interests
The authors declare no competing or financial interests.

### Author contributions
Conceptualization: M.K.I., L.C.K.; Data curation: M.K.I.; Formal analysis: M.K.I., A.K.S., J.v.S., H.N.V., B.C.D.; Funding acquisition: L.C.K.; Investigation: M.K.I., A.K.S., J.v.S., H.N.V., B.C.D.; Methodology: M.K.I., A.K.S.; Project administration: L.C.K.; Resources: L.C.K.; Software: M.K.I., H.N.V.; Supervision: L.C.K.; Visualization: M.K.I., A.K.S., H.N.V.; Writing – original draft: M.K.I., L.C.K.; Writing – review & editing: M.K.I., A.K.S., H.N.V., B.C.D., L.C.K.

### Funding
This work was supported by the European Research Council (consolidator grant 819219 – CellularLogistics to L.C.K.) and by the Dutch Foundation for Scientific Research (NOW-VICI grant VI.C.212.062 to L.C.K.). Open access funding provided by Utrecht University. Deposited in PMC for immediate release.

### Data and resource availability
Code used for motor-PAINT analysis has been deposited at GitHub (https://github.com/UU-cellbiology/neuron_motorPAINT). All other relevant data and details of resources can be found within the article and its supplementary information.

### First Person
This article has an associated First Person interview with the first author of the paper.

### Peer review history
The peer review history is available online at https://journals.biologists.com/jcs/lookup/doi/10.1242/jcs.264152.reviewer-comments.pdf

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
