## [Peer Review File · Journal of Cell Science]

Polarity reversal of stable microtubules during neuronal development

Malina K. Iwanski, Albert K. Serweta, Jasper Van Schelt, H. Noor Verwei, Bronte C. Donders and Lukas C. Kapitein
DOI: 10.1242/jcs.264152

Editor: Michael Way

Review timeline

Submission to Review Commons:	12 February 2025
Submission to Journal of Cell Science:	20 May 2025
Editorial decision:	21 May 2025
First revision received:	1 October 2025
Accepted:	8 October 2025

Reviewer 1

Evidence, reproducibility and clarity

This study examines the reorganization of the microtubule (MT) cytoskeleton during early neuronal development, specifically focusing on the establishment of axonal and dendritic polarity. Utilizing advanced microscopy techniques, the authors demonstrate that stable microtubules in early neurites initially exhibit a plus-end-out orientation, attributed to their connection with centrioles. Subsequently, these microtubules are released and undergo sliding, resulting in a mixed-polarity orientation in early neurites. Furthermore, the study elegantly illustrates the spatial segregation of microtubules in dendrites based on polarity and stability. The experiments are rigorously executed, and the microscopy data are presented with exceptional clarity. The following are my primary concerns that warrant further consideration by the authors.

1. Potential Bias in the MotorPAINT Assay:

Kinesin-1 and kinesin-3 motors exhibit distinct preferences for post-translationally modified (PTM) microtubules. Given that kinesin-1 preferentially binds to acetylated microtubules over tyrosinated microtubules in the MotorPAINT assay, the potential for bias in the results arises. Have the authors explored the use of kinesin-3, which favors tyrosinated microtubules, to corroborate the observed microtubule polarity?

2. Axon-Like Neurites in Stage 2b Neurons:

The observation of axon-like neurites in Stage 2b neurons, characterized by an (almost) uniformly plus-end-out microtubule organization, is noteworthy. Have the authors confirmed this polarity using end-binding (EB) protein tracking (e.g., EB1, EB3) in Stage 2b neurons? Do these neurites display distinct morphological features, such as variations in width? Furthermore, do they consistently differentiate into axons when tracked over time using live-cell EB imaging, rather than the MotorPAINT assay? Could stable microtubule anchoring impede free sliding in these neurites or restrict sliding into them? Investigating microtubule sliding dynamics in these axon-like neurites would provide valuable insights.

3. Taxol and Microtubule Sliding:

Taxol-induced microtubule stabilization is known to induce the formation of multiple axons. Does taxol treatment diminish microtubule sliding and prevent polarity reversal in minor neurites, thereby facilitating their development into axons?

4. Sorting of Minus-End-Out Microtubules (MTs) in Developing Axons:

Traces of minus-end-out MTs are observed proximal to the soma in both Stage 2b axon-like neurites and Stage 3 developing axons (Figure S4). Does this indicate a clearance mechanism for misoriented MTs during development? If so, is this sorting mechanism specific to axons? Could dynein be involved? Pharmacological inhibition of dynein (e.g., ciliobrevin-D or dynarrestin) could assess whether blocking dynein disrupts uniform MT polarity and axon formation.

5. Impact of Kinesin-1 Rigor Mutants on MT Polarity and Dynamics:

Would the expression of kinesin-1 rigor mutants alter MT dynamics and polarity? Validation with alternative methods, such as microtubule photoconversion, would be beneficial.

6. Molecular Motors Driving MT Sliding:

Which specific motors drive MT sliding in the soma and neurites? If a motor drives minus-end-out MTs into neurites, it must be plus-end-directed. The discussion should clarify the polarity of the involved motors to strengthen the conclusions.

7. Stability of Centriole-Derived Microtubules:

Microtubules emanating from centrioles are typically young and dynamic. How do they acquire acetylation and stability at an early stage? Do centrioles exhibit active EB1/EB3 comets in Stage 1/2a neurons? If these microtubules are severed from centrioles, could knockdown of MT-severing proteins (e.g., Katanin, Spastin, Fidgetin) alter microtubule polarity during neuronal development? A brief discussion would be valuable.

Minor Points:

1. In Movies 3 and 4, please use arrowheads or pseudo-coloring to highlight microtubules detaching from specific points. In Movie 5, please mark the stable microtubule that rotates within the neurite

2. In Movies 3 and 4, please use arrowheads or pseudo-coloring to highlight microtubules detaching from specific points. In Movie 5, mark the stable microtubule that rotates within the same neurite and the microtubule that exits and enters another neurite in the opposite orientation. These annotations would enhance clarity."

3. The title states: 'Stable microtubules predominantly oriented minus-end-out in the minor neurites of Stage 2b and 3 neurons.' However, given that the minus-end-out percentage increases after nocodazole treatment but only reaches a median of 0.48, 'predominantly' may be an overstatement. Please consider rewording.

4. Please compare the StableMARK system with the K560Rigor-SunTag approach described by Tanenbaum et al. (2014). What are the advantages of StableMARK over the SunTag method?

5. Microscopy data (Movies 2, 3, and 4) show microtubule bundling with StableMARK labeling, which is absent in tubulin immunostaining. Could this be an artifact of ectopic StableMARK expression? If so, a brief note addressing this potential effect would be beneficial.

Significance

It is an important paper challenging established ideas of microtubule organization in neurons. It is important to the wide audience of cell and neurobiologists.

Reviewer 2

Evidence, reproducibility and clarity

The manuscript uses state-of-the-art microscopy (e.g. expansion microscopy, motorPAINT) to observe microtubule organization during early events of differentiation of cultured rat hippocampal neurons. The authors confirm previous work showing that microtubules in neurites and dendrites are of mixed polarity whereas they are of uniform plus-end-out polarity in axons.

They show that stable microtubules (labeled with antibody against acetylated tubulin) are located in the central region of neurite cross-section across all differentiation stages. They show that acetylated microtubules are associated with centrioles early in differentiation but less so at later stages. And they show that stable microtubules can move from one neurite to another, presumably by microtubule sliding.

Comments:

- I found the manuscript difficult to read. There are lots of "segregations" of microtubules occurring over these stages of neuronal differentiation: segregation between the center of a neurite and the outer edge with respect to neurite cross-section, segregation between the region proximal to the cell body and the region distal to the cell body, and segregation over time (stages). The authors don't do a good job of distinguishing these and reporting the major findings in a way that is clear and straightforward.
- The major focus is on microtubule changes between stages 2a and 2b. This is introduced in the text and in the methods but not reflected in Figure 1A which should serve as an orientation of what is to come. It would be helpful to move the information about stages to the main text and/or Figure 1A.
- For Figure 1, the conclusions are generally supported by the data with the exception of the data for stage 2b in 1D and 1H. The images in D and the line scan in H suggest that for stage 2b, minus-end-out are on one edge whereas the plus-end-out are on the other edge of the neurite cross-section. But this is only true for one region along this example neurite. If the white line in D was moved proximal or distal along the neurite, the line scan for stage 2b would look like those of stages 2a and 3.
- For Figure 2, I found it difficult to relate panels A-F to panels G-J. I recommend combining 2G-J with 3A-B for a separate figure focused on the orientation of stable microtubules across different stages.
- For Figure 3, it is difficult to reconcile the traces with the corresponding images - that is, there are many acetylated microtubules in the top view image that appear to contact centrioles but are not in the tracing. Perhaps the tracings would more accurately reflect the localization of the acetylated microtubules in the top view images if a stack of images was shown rather than the max projections. Or if the authors were to stain for CAMSAPs to identify non-centrosomal microtubules. I find the data unconvincing but I do believe their conclusion because it is consistent with published data in the field. The data need to be quantified.
- I have a major concern with the conclusions of Figure 4. Here the authors use StableMARK to argue that microtubules do not depolymerize in one neurite and then repolymerize in another neurite but rather can be moved (presumably by sliding) from one neurite to another. The problem is that StableMARK-decorated microtubules do not depolymerize. So yes, StableMARK-decorated microtubules can move from one neurite to another but that does not say anything about what normally happens to microtubules during neuronal differentiation. In addition, the text says that the focus on Figure 4 is on how microtubules change between stages 2a and 2b but data is only shown for stage 2b.
- The data are largely descriptive and it is of course important to first describe things before one can dive into mechanism. But most of the findings confirm previous work and new findings are limited to showing that e.g. microtubule segregation appears earlier than previously observed.
- Optional: It would be nice if the authors could investigate some potential mechanisms. For example, does knockdown or knockout of severing enzymes prevent the loss of centriolar microtubules shown in Figure 3? Does knockdown or knockout of kinesin-2 or EB1 prevent the reorientation of microtubules (Chen et al 2014)?
- Overall, the methods are presented in such a way that they can be reproduced. One exception is in the motor paint sample prep section: is it three washes for 1 min each or three washes over 1 min?

- No statistical analysis is provided. The spread of the data in the violin plots is very large and it is difficult to ascertain how strongly one should make conclusions based on different data spreads between different conditions.
- For Figure S5, the excluded data (axons and axon-like neurites) should also be shown.
- For the movies, it would be helpful to have the microtubule moving from one neurite to another identified in some way as it is difficult to tell what is going on.

Significance

A strength of the study is the state-of-the-art microscopy (e.g. expansion microscopy, motorPAINT) and its application to a classic experimental model (rat hippocampal neurons). The information will be useful to those interested in the details of neuronal differentiation. A limitation of the study is that it appears to mostly confirm previous findings in the field (microtubule segregation, loss of centriolar anchoring, microtubule sliding). The advance to the field is that the manuscript shows that these events occur earlier in differentiation than previously known.

Reviewer 13

Evidence, reproducibility and clarity

The study by Iwanski and colleagues explores the establishment of the specific organisation of the neuronal microtubule cytoskeleton during neuronal differentiation. They use cultures of dissociated primary hippocampal rat neurons as a model system, and apply the optimised motor-PAINT technology, expansion microscopy/immunofluorescence and live cell imaging to investigate the polarity establishment and the distribution of differentially modified microtubules during early development.

They show that in young neurons microtubules are of mixed polarity, but at this stage already the stable (acetylated) microtubules are preferentially oriented plus-end-out, and are connected to the centrioles. In later stages, the stable microtubules are released from the centrioles and reverse their orientation by moving around inside the cell body and the neurites.

Major comments:

- Overall, the conclusions are well supported by the presented data.
- What is the proportion of neurons with different types of neurites (axon-like, non-axon-like) in stage 2b? (middle paragraph page 5 and Fig 1E). Please provide a quantification. How was the quantification in Fig 2B-D-F done? Why do the curves all start at 0? Please provide a scheme explaining these measurements. Furthermore, the data in Fig 2B do not reflect the statement "the segregation (...) was less evident" than in later stages (top of page 6): while it is less evident than in stage 2b, it is extremely similar to stage 3. Please revise accordingly.
- It should be stressed in the text, that the modification-specific antibodies only detect modified microtubules. Thus, in figure 3, in the absence of total tubulin staining, it is possible that there are more microtubules than revealed with the anti-acetylated tubulin antibody. A possible explanation should be discussed.
- OPTIONAL: As discussed in the manuscript's discussion, testing some of the proposed mechanisms regulating microtubule cytoskeleton architecture in development (motors, crosslinkers, severing enzymes) would significantly increase the impact of this study. Exploring these phenomena in a more complex system (3D culture, brain explants) closer to the intricate character of the brain than the 2D dissociated neurons would be a real game-changer.

Minor comments:

- The experiments are conducted thoroughly, the figures are clearly presented (for minor comments, see below) and the manuscript is well and clearly written.
- It could be useful to write on each panel whether the images were obtained with expansion or motor-PAINT technique: the rendering of the figures is very similar, and despite the different colour scheme can be confusing.

Significance

- This manuscript provides insights into the establishment of the microtubule cytoskeleton architecture specific to highly polarised neurons. The imaging techniques used, improved from the ones published before (motor-PAINT: Kapitein lab in 2017, U-ExM: Hamel/Guichard lab in 2019), yield beautiful and convincing data, marking an improvement compared to previous studies.
- However, the novelty of some of the findings is relatively limited. Indeed, a mixed microtubule orientation in very young neurites has already been shown (Yau et al, 2016, co- authored by Kapitein), as has the separate distribution of acetylated and tyrosinated / stable and labile / plus-end-out and plus-end-in microtubules in dendrites (Tas, ..., Kapitein, 2017).
- On the other hand, observation of the live movement of microtubules with the resolution allowing to see single (stable) microtubules is new and important. It provides an exciting setup to explore the mechanisms of polarity reversal of microtubules in neuronal development and it is regrettable that these mechanisms have not been explored further.
- The association of (stable) microtubules with the centrioles is also a technically challenging analysis. Despite not being able to visualise all microtubules, but only acetylated ones, these data are novel and exciting.
- This work will be of interest for neuronal cell biologists, developmental neurobiologists. The impact would be larger if the mechanistic questions were addressed using these sophisticated methodologies.
- This reviewer's expertise is the regulation of the microtubule cytoskeleton and it's impact on molecular, cellular and organism levels.

Author response to reviewers' comments

Manuscript number: RC-2025-02911
Corresponding author(s): Lukas Kapitein

[The “revision plan” should delineate the revisions that authors intend to carry out in response to the points raised by the referees. It also provides the authors with the opportunity to explain their view of the paper and of the referee reports.]

The document is important for the editors of affiliate journals when they make a first decision on the transferred manuscript. It will also be useful to readers of the reprint and help them to obtain a balanced view of the paper.

If you wish to submit a full revision, please use our "Full Revision" template. It is important to use the appropriate template to clearly inform the editors of your intentions.]

1. General Statements [optional]

This section is optional. Insert here any general statements you wish to make about the goal of the

study or about the reviews.

We are grateful to the reviewers for their thoughtful comments. Overall, all reviewers are positive about the study, in particular reviewers 1 and 3. Most comments can be addressed by further clarification or by adjustments to the text or figures. There are two key comments that are more substantive: 1/ to which extent does StableMARK overexpression result in artificially long microtubule lifetimes? 2/ the study is mostly descriptive and lacks insight into the mechanisms that drive microtubule reorganization.

To address the first comment, we will perform additional experiments using photoactivatable tubulin to examine microtubule lifetimes in neurons that do not overexpress StableMARK. We will also explain measures taken to minimize the risk of artifacts due to StableMARK overexpression based on previous characterization of this tool (Jansen *et al.*, 2023). With respect to the second comment, we note that two reviewers explicitly mention that exploring mechanisms is optional, while the other reviewer suggests two drug treatments. In future work, we hope to unravel the mechanisms underlying microtubule reorganization. However, this will require many more experiments, as well as the recruitment and training of a new PhD student or postdoc, given that the first author has left the lab. Therefore, we feel that this falls outside the scope of the current work, which carefully maps the microtubule organization during neuronal development and demonstrates the active polarity reversal of stable microtubules during this process. Because also drug treatments have their limitations, it would be better to present such experiments in a follow-up paper that also explores more controlled perturbations (i.e. knockdowns, inducible degradation etc.).

2. Description of the planned revisions

Insert here a point-by-point reply that explains what revisions, additional experimentations and analyses are planned to address the points raised by the referees.

Below, we summarize all points that were raised and the way we plan to address them. In the submission portal we have also uploaded the detailed point-to-point response, to be appended to our BioRxiv manuscript.

Summary of Reviewer 1

Reviewer 1 is very positive about our study: *“An important paper challenging established ideas of microtubule organization in neurons. It is important to the wide audience of cell and neurobiologists.” “The experiments are rigorously executed, and the microscopy data are presented with exceptional clarity”.*

Comments:

1. Does the subset selectivity of Kinesin-1 affect the motorPAINT experiments?
2. Can the uniform polarity in the axon-like neurites observed in stage 2b be validated using EB3 live imaging? Do these neurites develop into axons?
3. Can you examine how taxol treatment alters microtubules sliding?
4. Can you examine how dynein inhibition alters microtubule sliding?
5. Does expression of StableMARK cause microtubule overstabilization?
6. Can you discuss how motors could contribute to microtubule organization?
7. Can you discuss how centriole-attached microtubules become stabilized?

Planned revision:

1. Kinesin-1 is not selective in extracted cells. This will be clarified in the text.
2. We will perform EB3 experiments in stage 2b neurons. From earlier work by us and others we already know that this polarization is transient and can switch between neurites over time.
3. See below
4. See below
5. We will explain measures taken to limit the risk of artifacts and perform experiments using photoactivatable tubulin to test microtubule stability in the absence of StableMARK.
6. We will discuss this in the text

7. We will discuss this in the text

In future work, we hope to unravel the mechanisms underlying microtubule reorganization. However, this will require many more experiments, as well as the recruitment and training of a new PhD student or postdoc, given that the first author has left the lab. Therefore, we feel that this falls outside the scope of the current work, which carefully maps the microtubule organization during neuronal development and demonstrates the active polarity reversal of stable microtubules during this process. For this reason, we would prefer to leave out the experiment requested in questions 3 and 4.

Summary of Reviewer 2

Reviewer 2 is overall positive about our study: *“A strength of the study is the state-of-the-art microscopy (e.g. expansion microscopy, motorPAINT) and its application to a classic experimental model (rat hippocampal neurons). The information will be useful to those interested in the details of neuronal differentiation. A limitation of the study is that it appears to mostly confirm previous findings in the field (microtubule segregation, loss of centriolar anchoring, microtubule sliding). The advance to the field is that the manuscript shows that these events occur earlier in differentiation than previously known.”*

Comments:

1. Manuscript is difficult to read.
2. It would be helpful to move the information about stages to the main text and/or Figure 1A.
3. For Figure 1, the conclusions are generally supported by the data with the exception of the data for stage 2b in 1D and 1H.
4. I recommend combining 2G-J with 3A-B for a separate figure focused on the orientation of stable microtubules across different stages.
5. For Figure 3, it is difficult to reconcile the traces with the corresponding images - that is, there are many acetylated microtubules in the top view image that appear to contact centrioles but are not in the tracing.
6. StableMARK-decorated microtubules do not depolymerize. So yes, StableMARK-decorated microtubules can move from one neurite to another but that does not say anything about what normally happens to microtubules during neuronal differentiation.
7. The data are largely descriptive and it is of course important to first describe things before one can dive into mechanism.
8. Optional: It would be nice if the authors could investigate some potential mechanisms.
9. Motor paint sample prep section: is it three washes for 1 min each or three washes over 1 min?
10. No statistical analysis is provided.
11. For Figure S5, the excluded data (axons and axon-like neurites) should also be shown.
12. For the movies, it would be helpful to have the microtubule moving from one neurite to another identified in some way as it is difficult to tell what is going on.

Planned revision

To address comments 1, 2, 3, 4, 9-12, we will follow the reviewer's suggestions to improve readability and clarity of presentation.

To address comment 5, we will explain that the tracing was done using a 3D stack and that several microtubules that appear connected in the 2D projection are not in the 3D stack.

To address comment 6, we will perform photoactivation experiments to examine the lifetime of microtubules in cells that do not overexpress StableMARK.

Regarding comment 7 and 8: In future work, we hope to unravel the mechanisms underlying microtubule reorganization. However, this will require many more experiments, as well as the recruitment and training of a new PhD student or postdoc, given that the first author has left the lab. Therefore, we feel that this falls outside the scope of the current work, which carefully maps the microtubule organization during neuronal development and demonstrates the active polarity reversal of stable microtubules during this process.

Summary of reviewer 3

Reviewer is very positive about our study: *“Overall, the conclusions are well supported by the*

presented data. The experiments are conducted thoroughly, the figures are clearly presented (for minor comments, see below) and the manuscript is well and clearly written.” “Observation of the live movement of microtubules with the resolution allowing to see single (stable) microtubules is new and important.” “This work will be of interest for neuronal cell biologists, developmental neurobiologists.”

Comments:

1. Stage 2b neurites: provide quantification and clarification
2. Explain that modification-specific antibodies only detect modified microtubules
3. Optional: look into potential mechanism of microtubule reorganization

Planned revision:

Comments 1 and 2 can be addressed by providing additional quantifications and clarification. Regarding comment 3: In future work, we hope to unravel the mechanisms underlying microtubule reorganization. However, this will require many more experiments, as well as the recruitment and training of a new PhD student or postdoc, given that the first author has left the lab. Therefore, we feel that this falls outside the scope of the current work, which carefully maps the microtubule organization during neuronal development and demonstrates the active polarity reversal of stable microtubules during this process.

3. Description of the revisions that have already been incorporated in the transferred manuscript

Please insert a point-by-point reply describing the revisions that were already carried out and included in the transferred manuscript. If no revisions have been carried out yet, please leave this section empty.

N.A.

4. Description of analyses that authors prefer not to carry out

Please include a point-by-point response explaining why some of the requested data or additional analyses might not be necessary or cannot be provided within the scope of a revision. This can be due to time or resource limitations or in case of disagreement about the necessity of such additional data given the scope of the study. Please leave empty if not applicable.

As explained above, we understand the reviewers' suggestions to explore the mechanisms underlying the microtubule reorganization that we describe. In future work, we indeed hope to unravel the mechanisms underlying microtubule reorganization. However, this will require many more experiments, as well as the recruitment and training of a new PhD student or postdoc, given that the first author has left the lab. Therefore, we feel that this falls outside the scope of the current work, which carefully maps the microtubule organization during neuronal development and demonstrates the active polarity reversal of stable microtubules during this process. Specifically, at this point we would prefer to not explore the effect of taxol or dynein inhibitors on microtubule sliding (Reviewer 1, comments 3 and 4). Because also drug treatments have their limitations, it would be better to present such experiments in a follow-up paper that also explores more controlled perturbations (i.e. knockdowns, inducible degradation etc.).

Original submission

First decision letter

MS ID#: jcs.264152

MS TITLE: Polarity reversal of stable microtubules during neuronal development

AUTHORS: Malina K Iwanski; Albert K Serweta; H. Noor Verwei; Bronte C Donders; Lukas C Kapitein
 ARTICLE TYPE: Review Commons Transfer

Dear Lukas,

Thank you for submitting your paper that went through the review commons process. I think your responses and proposed revisions to address the three reviewers comments are appropriate. I would therefore be pleased to see a revised manuscript and will make a decision without going back to the reviewers once you return the manuscript.

First revision

Author response to reviewers' comments

Reviewer #1

(Evidence, reproducibility and clarity)

This study examines the reorganization of the microtubule (MT) cytoskeleton during early neuronal development, specifically focusing on the establishment of axonal and dendritic polarity. Utilizing advanced microscopy techniques, the authors demonstrate that stable microtubules in early neurites initially exhibit a plus-end-out orientation, attributed to their connection with centrioles. Subsequently, these microtubules are released and undergo sliding, resulting in a mixed-polarity orientation in early neurites. Furthermore, the study elegantly illustrates the spatial segregation of microtubules in dendrites based on polarity and stability. The experiments are rigorously executed, and the microscopy data are presented with exceptional clarity. The following are my primary concerns that warrant further consideration by the authors.

1. *Potential Bias in the MotorPAINT Assay: Kinesin-1 and kinesin-3 motors exhibit distinct preferences for post-translationally modified (PTM) microtubules. Given that kinesin-1 preferentially binds to acetylated microtubules over tyrosinated microtubules in the MotorPAINT assay, the potential for bias in the results arises. Have the authors explored the use of kinesin-3, which favors tyrosinated microtubules, to corroborate the observed microtubule polarity?*

We thank the reviewer for the careful assessment of our manuscript. As the reviewer noted, it has indeed been demonstrated that kinesin-1 prefers microtubules marked by acetylation (Cai et al., PLoS Biol 2009; Reed et al., Curr Biol 2006) and kinesin-3 prefers microtubules marked by tyrosination in cells (Guedes- Dias et al., Curr Biol 2019; Tas et al., Neuron 2017); however, these preferences are limited *in vitro*, as demonstrated for example in Sirajuddin et al. (Nat Cell Biol 2014). When motor-PAINT was introduced, it was verified that purified kinesin-1 moves over both acetylated and tyrosinated microtubules with no apparent preference in this assay (Tas et al., Neuron 2017). This could be due to the more *in vitro*-like nature of the motor-PAINT assay (e.g. some MAPs may be washed away) and/or because of the addition of Taxol during the gentle fixation step, which is known to alter the microtubule lattice. We have clarified this in the text by adding the following line: “Importantly, while kinesin-1 has been shown to selectively move on stable, highly modified microtubules in cells (Cai et al., 2009; Reed et al., 2006), this is not the case after motor-PAINT sample preparation (Tas et al., 2017)”

2. *Axon-Like Neurites in Stage 2b Neurons: The observation of axon-like neurites in Stage 2b neurons, characterized by an (almost) uniformly plus-end-out microtubule organization, is noteworthy. Have the authors confirmed this polarity using end-binding (EB) protein tracking (e.g., EB1, EB3) in Stage 2b neurons? Do these neurites display distinct morphological features, such as variations in width? Furthermore, do they consistently differentiate into axons when tracked over time using live-cell EB imaging, rather than the MotorPAINT assay? Could stable microtubule anchoring impede free sliding in these neurites or restrict sliding into*

them? Investigating microtubule sliding dynamics in these axon-like neurites would provide valuable insights.

We thank the reviewer for highlighting this finding. Early in development, cultured neurons are known to transiently polarize and have axon-like neurites that may or may not develop into the future axon (Burute et al., *Sci Adv* 2022; Schelski & Bradke, *Sci Adv* 2022; Jacobson et al., *Neuron* 2006). In the absence of certain molecular or physical factors (e.g. Burute et al., *Sci Adv* 2022; Randlett et al., *Neuron* 2011), this transient polarization is seemingly random and as such, we do not expect the axon-like neurites in stage 2b neurons to necessarily become the axon. Interestingly, anchoring stable microtubules in a specific neurite using cortically-anchored StableMARK (Burute et al., *Sci Adv* 2022) or stabilizing microtubules in a specific neurite using Taxol (Witte et al., *JCB* 2008) has been shown to promote axon formation, but these stable microtubules have slower turnover (perhaps necessitating the use of laser severing as in Yau et al., *J Neurosci* 2016) and may not always bear EB comets given that EB comets are less commonly seen at the ends of stable microtubules (Jansen et al., *JCB* 2023).

Upon request of the reviewer, we have performed EB3 tracking experiments in Stage 2b neurons to examine potential differences between neurites. We found that in all neurites, EB3 comets were mostly moving anterogradely, consistent with our earlier published results (Yau et al., *J Neurosci* 2016). Because of the similarity with previously published work, we decided not to include this data into this manuscript.

3. *Taxol and Microtubule Sliding: Taxol-induced microtubule stabilization is known to induce the formation of multiple axons. Does taxol treatment diminish microtubule sliding and prevent polarity reversal in minor neurites, thereby facilitating their development into axons?*

We are very grateful for this interesting suggestion. In the revised manuscript, we now include motor-PAINT experiments showing that the multiple axons induced by Taxol treatment (10 nM) all contain predominantly plus-end-out microtubules. Furthermore, new live-cell experiments confirm the suspicion of the reviewer that even at these low concentration Taxol readily diminishes microtubule sliding and thereby impedes polarity reversal.

4. *Sorting of Minus-End-Out Microtubules (MTs) in Developing Axons: Traces of minus-end-out MTs are observed proximal to the soma in both Stage 2b axon-like neurites and Stage 3 developing axons (Figure S4). Does this indicate a clearance mechanism for misoriented MTs during development? If so, is this sorting mechanism specific to axons? Could dynein be involved? Pharmacological inhibition of dynein (e.g., ciliobrevin-D or dynarrestin) could assess whether blocking dynein disrupts uniform MT polarity and axon formation.*

We indeed think that a clearance mechanism is involved for removing misoriented microtubules in the axon after axon specification. Many motor proteins have been implicated in the polarity sorting of microtubules in neurons and for axons, dynein is believed to play a role (Rao et al., *Cell Rep* 2017; del Castillo et al., *eLife* 2015; Schelski & Bradke, *Sci Adv* 2022). A few of these studies already employed ciliobrevin, noting that it increases the fraction of minus-end-out microtubules in axons (Rao et al., *Cell Rep* 2017) and reduces the rate of retrograde flow of microtubules in immature neurites (Schelski & Bradke, *Sci Adv* 2022). These findings are in line with the suggestion of the reviewer. Interestingly, however, as we highlight in the discussion, the motility we observe for polarity reversal is extremely slow on average (~60 nm/minute) because the microtubule end undergoes bursts of motility and periods in which it appears to be tethered and rather immobile. Given that most neurites are non-axon-like, we assume these sliding events are mostly not taking place in axons or axon-like neurites. These events may thus be orchestrated by other motor proteins (e.g. kinesin-1, kinesin-2, kinesin-5, kinesin-6, and kinesin-12) that have been implicated in microtubule polarity sorting in neurons. We do observe retrograde sliding of stable microtubules in these neurites at a median speed of ~150 nm/minute, which is again much slower than typical motor speeds and occurs in almost all neurites and not specifically in one or two axon-like neurites. It is thus unclear which motors may be involved, and it is difficult to predict how any drug treatments would affect microtubule polarity.

While we agree that more obtaining more mechanistic understanding is an exciting goal for future work, we have agreed with the editor that this fall outside the scope of the current work,

which carefully maps the microtubule organization during neuronal development and demonstrates the active polarity reversal of stable microtubules during this process. In the discussion section, we discuss potential mechanisms facilitating polarity sorting in axons and axon-like neurites.

5. *Impact of Kinesin-1 Rigor Mutants on MT Polarity and Dynamics: Would the expression of kinesin-1 rigor mutants alter MT dynamics and polarity? Validation with alternative methods, such as microtubule photoconversion, would be beneficial.*

It is important to note that StableMARK and its effects on microtubule stability have been extensively verified in the paper in which it was introduced (Jansen et al., JCB 2023). At low expression levels (where StableMARK has a speckled distribution along microtubules), StableMARK does not alter the stability of microtubules (e.g., they are still disassembled in response to serum starvation), alter their post-translational modification status or their distribution in the cell, or impede the transport of cargoes along them. Given that we chose to image neurons with very low expression levels of StableMARK (as inferred by the speckled distribution along microtubules), we expect its effects on the microtubule cytoskeleton to be minimal.

To provide further evidence for the existence of long-lived microtubules in developing neurons, we have followed the reviewer's suggestion and used photoactivatable tubulin in cells without StableMARK to examine whether we still see microtubules that live for over 2 hours. These data are shown in Figure S5A-B and explained in the main text as follows: "We independently verified this long lifetime of a sub-population of microtubules using photoactivatable tubulin. This approach has previously been used to observe the motile behaviour of microtubules over time (Burute et al., 2022; Lu et al., 2016). Photoactivated tubulin not incorporated into microtubules is expected to quickly dissipate. Instead, tubulin incorporated into persistent microtubules provides a persistent signal. We could indeed confirm the continued presence of signal after two or more hours (Figure S5A-B)."

6. *Molecular Motors Driving MT Sliding: Which specific motors drive MT sliding in the soma and neurites? If a motor drives minus-end-out MTs into neurites, it must be plus-end-directed. The discussion should clarify the polarity of the involved motors to strengthen the conclusions.*

We thank the reviewer for highlighting this point and have improved our discussion to clarify the polarity of the involved motors.

7. *Stability of Centriole-Derived Microtubules: Microtubules emanating from centrioles are typically young and dynamic. How do they acquire acetylation and stability at an early stage? Do centrioles exhibit active EB1/EB3 comets in Stage 1/2a neurons? If these microtubules are severed from centrioles, could knockdown of MT-severing proteins (e.g., Katanin, Spastin, Fidgetin) alter microtubule polarity during neuronal development? A brief discussion would be valuable.*

We thank the reviewer for raising these interesting questions and suggestions. As suggested, we will include a brief discussion of these issues. What is known about the properties of stable microtubules is limited, so it is currently unclear how they are made. For example, we do not know if they are converted from labile microtubules or nucleated by a distinct pathway. If they are nucleated by a distinct pathway, do these microtubules grow in a similar manner as labile microtubules and do they have EB comets at their plus-ends (given that EB compacts the lattice (Zhang et al., Cell 2015, PNAS 2018) and stable microtubules have an expanded lattice in cells (de Jager et al., JCB 2025))? If they are converted, does something first cap their plus-end to limit further growth (given that EB comets are rarely observed at the ends of stable microtubules (Jansen et al., JCB 2023))?

We also do not know how the activity of the tubulin acetyltransferase α TAT1 is regulated. Is its access to the microtubule lumen regulated or is its enzymatic activity stimulated by some means (e.g., microtubule lattice conformation or a molecular factor)?

We find the possibility that microtubule severing enzymes release these stable microtubules from the centrioles very exciting and hope to test the effects of their absence on microtubule polarity

in the future. We will discuss this in the manuscript as suggested.

Minor Points

1. *In Movies 3 and 4, please use arrowheads or pseudo-coloring to highlight microtubules detaching from specific points. In Movie 5, please mark the stable microtubule that rotates within the neurite. These annotations would enhance clarity.*

Movies 3 and 4 show the general dynamics of stable microtubules in developing neurons, but the density of microtubules prevents us from following the dynamics of specific microtubules. For Movie 5, we have now added traces to the movie to enhance clarity.

2. *The title states: 'Stable microtubules predominantly oriented minus-end-out in the minor neurites of Stage 2b and 3 neurons.' However, given that the minus-end-out percentage increases after nocodazole treatment but only reaches a median of 0.48, 'predominantly' may be an overstatement. Please consider rewording.*

We have adjusted the statement to better reflect the median value.

3. *Please compare the StableMARK system with the K560Rigor-SunTag approach described by Tanenbaum et al. (2014). What are the advantages of StableMARK over the SunTag method?*

While the SunTag is certainly a powerful tool to visualize molecules at low copy number, we believe that StableMARK is more appropriate than the K560Rigor-SunTag tool for our assays due to two main reasons.

Firstly, K560Rigor-SunTag is based on the E236A kinesin-1 mutation, while StableMARK is based on the G234A mutation. These are both rigor mutations of kinesin-1 but behave differently; the E236A mutant is strongly bound to the microtubule in an ATP-like state (neck linker docked), while the G234A mutant is also strongly bound, but not in an ATP-like state (Rice et al., Nature 1999). This means that they may have different effects on or preferences of the microtubule lattice. Indeed, while StableMARK (G234A) has been shown to preferentially bind microtubules with an expanded lattice (Jansen et al., JCB 2023; de Jager et al., JCB 2025), this may not be the case for the E236A mutant. In support of this, it has been shown that, while nucleotide free kinesin-1 can expand the lattice of GDP-microtubules at high concentrations (>10% lattice occupancy) in vitro (Peet et al., Nat Nanotechnol 2018; Shima et al., JCB 2018), kinesin-1 in the ATP-bound state does not maintain this expanded lattice (Shima et al., JCB 2018). Thus, we expect the kinesin-1 rigor used by Tanenbaum et al. (Cell 2014) to not be specific for stable microtubules (with an expanded lattice) in cells. In addition, given the dense packing of microtubules in neurites (not well-established in developing neurites, but with an inter-microtubule distance of ~25 nm in axons and ~65 nm in dendrites (Chen et al., Nature 1992)), the very large size of the SunTag could be problematic. The K560Rigor-SunTag tool from Tanenbaum et al. (Cell 2014) is bound by up to 24 copies of GFP (each ~3 nm in size), meaning that it may obstruct or be obstructed by the dense microtubule network in neurites.

Given that, unlike the K560Rigor-SunTag construct, StableMARK has been carefully validated as a live- cell marker for stable microtubules, we believe that the above discussion goes beyond the scope of the manuscript.

4. *Microscopy data (Movies 2, 3, and 4) show microtubule bundling with StableMARK labeling, which is absent in tubulin immunostaining. Could this be an artifact of ectopic StableMARK expression? If so, a brief note addressing this potential effect would be beneficial.*

As with any overexpression, there is a risk of artifacts. We feel that in the cells presented, the risk of artifacts is limited because we have chosen neurons expressing StableMARK at very low levels. Prior work has demonstrated that in cells where StableMARK has a speckled appearance on microtubules, it has limited undesired effects on stable microtubules or the cargoes moving along them (Jansen et al., JCB 2023). Perhaps some of the apparent differences in the amount of bundling can be explained in that the expansion microscopy images shown may have less apparent bundling because of the improved z-resolution and thus optical sectioning. Any z-slice imaged

using expansion microscopy will contain fewer microtubules, so bundling may be less obvious. If we compare the amount of bundling seen in StableMARK expressing cells with the amount of bundling of acetylated microtubules (a marker for stable microtubules) in DMSO/nocodazole treated (non-electroporated) cells imaged by confocal microscopy in Figure S7, we feel that the difference is not so large.

Reviewer #1

(Significance)

It is an important paper challenging established ideas of microtubule organization in neurons. It is important to the wide audience of cell and neurobiologists.

Reviewer #2

(Evidence, reproducibility and clarity)

The manuscript uses state-of-the-art microscopy (e.g. expansion microscopy, motorPAINT) to observe microtubule organization during early events of differentiation of cultured rat hippocampal neurons. The authors confirm previous work showing that microtubules in neurites and dendrites are of mixed polarity whereas they are of uniform plus-end-out polarity in axons. They show that stable microtubules (labeled with antibody against acetylated tubulin) are located in the central region of neurite cross-section across all differentiation stages. They show that acetylated microtubules are associated with centrioles early in differentiation but less so at later stages. And they show that stable microtubules can move from one neurite to another, presumably by microtubule sliding.

Comments

- 1. I found the manuscript difficult to read. There are lots of "segregations" of microtubules occurring over these stages of neuronal differentiation: segregation between the center of a neurite and the outer edge with respect to neurite cross-section, segregation between the region proximal to the cell body and the region distal to the cell body, and segregation over time (stages). The authors don't do a good job of distinguishing these and reporting the major findings in a way that is clear and straightforward.*

We thank the reviewer for their feedback and will go over the text to make it easier to read. Within neurites, we use the word 'segregated' in the manuscript to mean that the microtubules form two spatially separate populations across the width of the neurites (i.e., their cross-section if viewed in 3D). Because of variability seen in the neurites of this stage, this segregation does not always present as a peripheral vs. central enrichment of the different populations of microtubules as we sometimes observed two side-by-side populations instead. We will make sure that we properly define this in the manuscript to avoid any confusion.

When discussing other types of segregation, we tried to use different wording such as when discussing the proximal-distal distribution of microtubules with different orientations in axon-like neurites in this excerpt:

Sometimes these axons and axon-like neurites had a small bundle of minus-end-out microtubules proximal to the soma (Figure S4). This suggests that plus-end-out uniformity emerges distally first in these neurites, perhaps by retrograde sliding of these minus-end-out microtubules (see Discussion).

When discussing changes related to a particular stage, we instead aimed to list which stage we were talking about, such as seen in the discussion:

Emerging neurites of early stage 2 neurons already contain microtubules of both orientations and these are typically segregated. These emerging neurites also contain segregated networks of acetylated (stable) and tyrosinated (labile) microtubules. In later stage 2, stage 3, and stage 4 neurons, stable (nocodazole-resistant) microtubules are oriented more minus-end-out compared to

the total (untreated) population of microtubules; however, in early stage 2 neurons, stable microtubules are preferentially oriented plus-end-out, likely because their minus-ends are still anchored at the centrioles at this stage. The fraction of anchored stable microtubules decreases during development, while the appearance of short stumps of microtubules attached to the centrioles suggests that these microtubules may be released by severing.

We have reviewed the text carefully for clarity.

- 2. The major focus is on microtubule changes between stages 2a and 2b. This is introduced in the text and in the methods but not reflected in Figure 1A which should serve as an orientation of what is to come. It would be helpful to move the information about stages to the main text and/or Figure 1A.*

We thank the reviewer for pointing this out and are now more explicit about the distinction between stages 2a and 2b in the main text and have made the suggested change to Figure 1A.

- 3. For Figure 1, the conclusions are generally supported by the data with the exception of the data for stage 2b in 1D and 1H. The images in D and the line scan in H suggest that for stage 2b, minus-end-out are on one edge whereas the plus-end-out are on the other edge of the neurite cross-section. But this is only true for one region along this example neurite. If the white line in D was moved proximal or distal along the neurite, the line scan for stage 2b would look like those of stages 2a and 3.*

We thank the reviewer for noting this in the figure. For these earlier stages in neuronal development, the distribution of different types of microtubules within the neurite is more variable and does not always adhere to the central-peripheral distribution described for more mature neurons (Tas et al., Neuron 2017). We did not intend to suggest that neurites of stage 2b neurons consistently have a different radial distribution of microtubules of opposite orientation, but rather that microtubules of the same orientation tend to bundle together. Sometimes this bundling produces a central or peripheral enrichment, as described for mature neurons (Tas et al., Neuron 2017) and as seen in Figure 1D-F at certain points along the length of the neurites, and sometimes the bundling simply produces two side-by-side populations. To reflect this diversity, we chose two different examples in the figure. The line scans presented in Figure 1H were taken approximately at the midpoint of the presented ROIs. In addition, as our imaging in this case is two-dimensional, we do not want to make explicit claims about the radial distribution of the different populations of microtubules.

In the revised manuscript, we have adjusted our description of this figure to be more explicit about how we interpret these results. We tried to ensure that it is apparent that we do not think there is a specific radial distribution of microtubules depending on the developmental stage.

- 4. For Figure 2, I found it difficult to relate panels A-F to panels G-J. I recommend combining 2G-J with 3A-B for a separate figure focused on the orientation of stable microtubules across different stages.*

We thank the reviewer for this suggestion and now present this data in two separate figures.

- 5. For Figure 3, it is difficult to reconcile the traces with the corresponding images - that is, there are many acetylated microtubules in the top view image that appear to contact centrioles but are not in the tracing. Perhaps the tracings would more accurately reflect the localization of the acetylated microtubules in the top view images if a stack of images was shown rather than the max projections. Or if the authors were to stain for CAMSAPs to identify non-centrosomal microtubules. I find the data unconvincing but I do believe their conclusion because it is consistent with published data in the field. The data need to be quantified.*

We thank the reviewer for noting this. Importantly, the tracing was done on a three-dimensional stack of images, whereas we present maximum projections of a few slices in Figure 3C for easy visualization. Projection artifacts indeed make it look as though some additional microtubules are attached to the centrioles, whereas in the three-dimensional stacks it is apparent that they are not. We now clarify this in the method section related to Figure 3C. We now also include the z-

stacks as supplementary material so that readers can also verify this themselves.

6. *I have a major concern with the conclusions of Figure 4. Here the authors use StableMARK to argue that microtubules do not depolymerize in one neurite and then repolymerize in another neurite but rather can be moved (presumably by sliding) from one neurite to another. The problem is that StableMARK-decorated microtubules do not depolymerize. So yes, StableMARK-decorated microtubules can move from one neurite to another but that does not say anything about what normally happens to microtubules during neuronal differentiation. In addition, the text says that the focus on Figure 4 is on how microtubules change between stages 2a and 2b but data is only shown for stage 2b.*

As noted by the reviewer, StableMARK can indeed hyperstabilize microtubules when over-expressed; however, it is important to note that this strongly depends on the level of overexpression of the marker. This is discussed in detail in the paper introducing StableMARK, where it is described that at low expression levels, StableMARK does not alter the stability of microtubules (i.e., StableMARK decorated microtubules can still depolymerize/disassemble and they are disassembled in response to serum starvation), alter their post-translational modification status or their distribution in the cell, or impede the transport of cargoes along them (Jansen et al. JCB 2023).

To provide independent evidence for the existence of long-lived microtubules in developing neurons, we have now used photoactivatable tubulin in cells without StableMARK to examine whether we still see microtubules that live for over 2 hours. These data are shown in Figure S5A-B and explained in the main text as follows: “We independently verified this long lifetime of a sub-population of microtubules using photoactivatable tubulin. This approach has previously been used to observe the motile behaviour of microtubules over time (Burute et al., 2022; Lu et al., 2016). Photoactivated tubulin not incorporated into microtubules is expected to quickly dissipate. Instead, tubulin incorporated into persistent microtubules provides a persistent signal. We could indeed confirm the continued presence of signal after two or more hours (Figure S5A-B).”

7. *The data are largely descriptive and it is of course important to first describe things before one can dive into mechanism. But most of the findings confirm previous work and new findings are limited to showing that e.g. microtubule segregation appears earlier than previously observed.*

Our study is the first to use Motor-PAINT to carefully map changes in microtubule orientations during neuronal development. Furthermore, it is the first to use the recently introduced live-cell marker for stable microtubules to directly demonstrate the active polarity reversal of stable microtubules during this process.

8. *Optional: It would be nice if the authors could investigate some potential mechanisms. For example, does knockdown or knockout of severing enzymes prevent the loss of centriolar microtubules shown in Figure 3? Does knockdown or knockout of kinesin-2 or EB1 prevent the reorientation of microtubules (Chen et al 2014)?*

While we agree that more obtaining more mechanistic understanding is an exciting goal for future work, we have agreed with the editor that this fall outside the scope of the current work, which carefully maps the microtubule organization during neuronal development and demonstrates the active polarity reversal of stable microtubules during this process.

9. *Overall, the methods are presented in such a way that they can be reproduced. One exception is in the motor paint sample prep section: is it three washes for 1 min each or three washes over 1 min?*

We thank the reviewer for pointing out this mistake and have adjusted this step in the methods section accordingly.

10. *No statistical analysis is provided. The spread of the data in the violin plots is very large and it is difficult to ascertain how strongly one should make conclusions based on different data spreads between different conditions.*

In the revised manuscript, we have included statistical tests in the specified graphs.

11. *For Figure S5, the excluded data (axons and axon-like neurites) should also be shown.*

For this figure, the excluded data points are simply those neurites with <10% minus-end-out localizations. We have now clearly specified this in the graph and figure legend, but do not feel that it warrants the addition of another graph. We have also reorganized the supplementary figures to facilitate the comparison with other relevant data (e.g., the orientation in the longest neurite and the orientation after Taxol treatment).

12. *For the movies, it would be helpful to have the microtubule moving from one neurite to another identified in some way as it is difficult to tell what is going on.*

We have traced the microtubule in this movie to enhance clarity.

Reviewer #2

(Significance)

A strength of the study is the state-of-the-art microscopy (e.g. expansion microscopy, motorPAINT) and its application to a classic experimental model (rat hippocampal neurons). The information will be useful to those interested in the details of neuronal differentiation. A limitation of the study is that it appears to mostly confirm previous findings in the field (microtubule segregation, loss of centriolar anchoring, microtubule sliding). The advance to the field is that the manuscript shows that these events occur earlier in differentiation than previously known.

Reviewer #3

(Evidence, reproducibility and clarity)

The study by Iwanski and colleagues explores the establishment of the specific organisation of the neuronal microtubule cytoskeleton during neuronal differentiation. They use cultures of dissociated primary hippocampal rat neurons as a model system, and apply the optimised motor-PAINT technology, expansion microscopy/immunofluorescence and live cell imaging to investigate the polarity establishment and the distribution of differentially modified microtubules during early development.

They show that in young neurons microtubules are of mixed polarity, but at this stage already the stable (acetylated) microtubules are preferentially oriented plus-end-out, and are connected to the centrioles. In later stages, the stable microtubules are released from the centrioles and reverse their orientation by moving around inside the cell body and the neurites.

Overall, the conclusions are well supported by the presented data. The experiments are conducted thoroughly, the figures are clearly presented (for minor comments, see below) and the manuscript is well and clearly written.

Major comments

1. *What is the proportion of neurons with different types of neurites (axon-like, non-axon-like) in stage 2b? (middle paragraph page 5 and Fig 1E). Please provide a quantification. How was the quantification in Fig 2B-D-F done? Why do the curves all start at 0? Please provide a scheme explaining these measurements. Furthermore, the data in Fig 2B do not reflect the statement "the segregation (...) was less evident" than in later stages (top of page 6): while it is less evident than in stage 2b, it is extremely similar to stage 3. Please revise accordingly.*

We thank the reviewer for pointing out these important details. We have made the suggested changes in the text and added the proportion of neurons with different types of neurites.

The radial intensity distributions were quantified as described in Katrukha et al. (eLife 2021). In the methods section, we describe the process in brief:

To analyze the radial distribution of acetylated and tyrosinated microtubules in expanded neurites, deconvolved image stacks were processed using custom scripts in ImageJ (v1.54f) and MATLAB (R2024b) as described in detail elsewhere (Katrukha et al., 2021). Briefly, on maximum intensity projections (XY plane), we drew polylines of sufficient thickness (300 px) to segment out neurite portions 44 μm (10 μm when corrected for expansion factor) in length proximal to the cell soma. Using Selection > Straighten on the corresponding z-stacks generated straightened B-spline interpolated stacks of the neurite sections. These z-stacks were then resliced perpendicularly to the neurite axis (YZ-plane) to visualize the neurite cross-section. From this, we could semi-automatically find the boundary of the neurite in each slice using first a bounding rectangle that encompasses the neurite (per slice) and then a smooth closed spline (approximately oval). To build a radial intensity distribution from neurite border to center, closed spline contours were then shrunken pixel by pixel in each YZ- slice while measuring ROI area and integrated fluorescence intensity. From this, we could ascertain the average fluorescence intensity per contour iteration, allowing us to calculate a radial intensity distribution by calculating the radius corresponding to each area (assuming the neurite cross-section is circular).

The curves thus all start at 0 because no intensity “fits” into a circle of radius 0 and then gradually increase because very few microtubules “fit” into circles with the smallest radii. We added a brief statement to the methods section to explain why the curves start at 0.

- 2. It should be stressed in the text, that the modification-specific antibodies only detect modified microtubules. Thus, in figure 3, in the absence of total tubulin staining, it is possible that there are more microtubules than revealed with the anti-acetylated tubulin antibody. A possible explanation should be discussed.*

We thank the reviewer for highlighting this point and have adjusted the text accordingly.

- 3. OPTIONAL: As discussed in the manuscript's discussion, testing some of the proposed mechanisms regulating microtubule cytoskeleton architecture in development (motors, crosslinkers, severing enzymes) would significantly increase the impact of this study. Exploring these phenomena in a more complex system (3D culture, brain explants) closer to the intricate character of the brain than the 2D dissociated neurons would be a real game-changer.*

While we agree that more obtaining more mechanistic understanding is an exciting goal for future work, we have agreed with the editor that this fall outside the scope of the current work, which carefully maps the microtubule organization during neuronal development and demonstrates the active polarity reversal of stable microtubules during this process.

Minor comments

- 1. It could be useful to write on each panel whether the images were obtained with expansion or motor- PAINT technique: the rendering of the figures is very similar, and despite the different colour scheme can be confusing.*

To clarify this point, we now show the motor-PAINT and expansion data in different figures.

Reviewer #3

(Significance)

This manuscript provides insights into the establishment of the microtubule cytoskeleton architecture specific to highly polarised neurons. The imaging techniques used, improved from the ones published before (motor- PAINT: Kapitein lab in 2017, U-ExM: Hamel/Guichard lab in 2019), yield beautiful and convincing data, marking an improvement compared to previous studies.

However, the novelty of some of the findings is relatively limited. Indeed, a mixed microtubule orientation in very young neurites has already been shown (Yau et al, 2016, co-authored by Kapitein), as has the separate distribution of acetylated and tyrosinated / stable and labile / plus-end-out and plus-end-in microtubules in dendrites (Tas, ..., Kapitein, 2017).

On the other hand, observation of the live movement of microtubules with the resolution allowing to see single (stable) microtubules is new and important. It provides an exciting setup to explore the mechanisms of polarity reversal of microtubules in neuronal development and it is regrettable that these mechanisms have not been explored further.

The association of (stable) microtubules with the centrioles is also a technically challenging analysis. Despite not being able to visualise all microtubules, but only acetylated ones, these data are novel and exciting.

This work will be of interest for neuronal cell biologists, developmental neurobiologists. The impact would be larger if the mechanistic questions were addressed using these sophisticated methodologies.

This reviewer's expertise is the regulation of the microtubule cytoskeleton and its impact on molecular, cellular and organism levels.

Second decision letter

MS ID#: jcs.264152R1

MS Title: Polarity reversal of stable microtubules during neuronal development

Authors: Malina K Iwanski; Albert K Serweta; Jasper Van Schelt; H. Noor Verweij; Bronte C Donders; Lukas C Kapitein

Article Type: Review Commons Transfer

Dear Lukas,

Based on your revisions and new experiments, I am happy to tell you that your manuscript has been accepted for publication in Journal of Cell Science, pending standard publication integrity checks.

Thank you for sending your manuscript to Journal of Cell Science through Review Commons.